# DARE the Extreme 🏃: Revisiting Delta-Parameter Pruning For Fine-Tuned Models

**Wenlong Deng**[1,2]**, Yize Zhao**[1]**, Vala Vakilian**[1]**, Minghui Chen**[1,2]**, Xiaoxiao Li**[1,2]**,
Christos Thrampoulidis**[1]
[1]The University of British Columbia    [2]Vector Institute
`https://github.com/vengdeng/DAREx.git`

## ABSTRACT

Storing open-source fine-tuned models separately introduces redundancy and increases response times in applications utilizing multiple models. Delta-parameter pruning (DPP), particularly the random drop and rescale (DARE) method proposed by Yu et al., addresses this by pruning the majority of delta parameters—the differences between fine-tuned and pre-trained model weights—while typically maintaining minimal performance loss. However, DARE fails when either the pruning rate or the magnitude of the delta parameters is large. We highlight two key reasons for this failure: (1) an excessively large rescaling factor as pruning rates increase, and (2) high mean and variance in the delta parameters. To push DARE's limits, we introduce DAREx (DARE the eXtreme), which features two algorithmic improvements: (1) DAREx-q, a rescaling factor modification that significantly boosts performance at high pruning rates (e.g., $> 30\%$ on COLA and SST2 for encoder models, with even greater gains in decoder models), and (2) DAREx-$L_2$, which combines DARE with AdamR, an in-training method that applies appropriate delta regularization before DPP. We also demonstrate that DAREx-q can be seamlessly combined with vanilla parameter-efficient fine-tuning techniques like LoRA and can facilitate structural DPP. Additionally, we revisit the application of importance-based pruning techniques within DPP, demonstrating that they outperform random-based methods when delta parameters are large. Through this comprehensive study, we develop a pipeline for selecting the most appropriate DPP method under various practical scenarios.

## 1 INTRODUCTION

Large Language Models (LLMs) like BERT (Devlin et al., 2018), GPT (Floridi & Chiriatti, 2020), and Llama (Touvron et al., 2023) excel in various language-modeling tasks. Finetuning these for specific downstream tasks enhances personalized experiences (Sanh et al., 2021; Labrak et al., 2024) and has become a standard practice in natural language processing (Dodge et al., 2020; Zhao et al., 2023). Indeed, most entries on the OpenLLM Leaderboard involve full-parameter finetunes or their combinations (Liu et al., 2024), underscoring the widespread adoption and availability of fine-tuned models online. Decomposing fine-tuned model weights into the original parameters of the pre-trained model yields *delta parameters* (DP) (Yu et al., 2023a; Liu et al., 2024; Yao & Klimovic, 2023). Reducing the size of DPs, which are as large as the base model and can number in the hundreds of millions of parameters for LLMs, could significantly enhance communication efficiency in federated learning, minimize task conflicts in model merging, accelerate multi-task serving, and decrease storage needs for new fine-tuned models (see *Related Work*).

Delta-parameter pruning (DPP) drops a fraction $p$ of the DPs towards realizing these benefits. Naturally, DPP can be seen as an instance of generic model-parameter pruning, which compresses neural networks by eliminating weights that contribute minimally, resulting in sparse architectures (LeCun et al., 1989; Han

et al., 2015b). Traditional pruning methods typically remove weights post-training based on *importance criteria* like weight magnitude or activation levels. While these techniques could naturally extend to DPP, their integration into this context remains largely unexplored.

Random-based pruning strategies, which provide more flexibility and efficiency in implementation, also offer a competitive alternative. For instance, Random Drop and Rescale (DARE) (Yu et al., 2023a), a recently introduced randomized DPP method, reduces DP size through random pruning followed by rescaling. DARE has been quickly adopted across various applications, including model merging libraries (Goddard et al., 2024), state-of-the-art medical LLMs (Labrak et al., 2024), and Japanese LLMs (Akiba et al., 2024).

However, as we demonstrate in this paper, DARE struggles when the pruning rate is high or when DPs are large. This observation prompts several key questions:

> *What are the key factors contributing to DARE's failure modes? Can these issues be addressed to push the limits of effective random-based DPP? Additionally, can importance-based model pruning techniques be applied to DPP in ways that compete with random-based methods?*

## 1.1 CONTRIBUTIONS

We address these questions through principled analysis and extensive experimentation on large-scale language models and datasets. Our contributions are summarized as follows:

• **Analysis of DARE:** By examining the *absolute change in intermediate output model representations* resulting from the application of DARE to the DPs, we identify two primary factors that influence this change: **(a)** a large pruning rate ($p$), which results in an excessively high rescaling factor of $1/(1-p)$, and **(b)** a high mean and variance of the DPs relative to input activations. Through experiments on both controlled setups and LLMs, we show that the absolute change in intermediate representations is a reliable proxy for

| Pruning Rate | $p = 0$ | $p = 0.99$ |
|---|---|---|
| Delta Size | 417.7MB | **11.4MB** |
| Method | COLA | SST2 |
| ($p = 0.99$) | Test Performance (%) | Test Performance (%) |
| No Pruning | 56.24 | 90.25 |
| DARE (Yu et al., 2023a) | 4.25 | 51.00 |
| $L_1$+MP (Han et al., 2015a) (ours) | 12.30 (+8.05) | 83.14 (+32.14) |
| DAREx-$L_2$ (ours) | 57.24 (+52.99) | 88.17 (+37.00) |
| DAREx-q ($1/q_v$) (ours) | 48.96 (+44.71) | 85.64 (+34.64) |
| DAREx-q ($1/q_e$) (ours) | 45.20 (+41.05) | 85.81 (+34.81) |

Table 1: *Impact of our key contributions on BERT and two datasets COLA and SST2*: 'DAREx-$L_2$ ' applies DARE after AdamR-$L_2$ fine-tuning, '$L_1$+MP' applies magnitude pruning after AdamR-$L_1$, the factors $q_v$ and $q_e$ adjust DARE's rescaling using a validation set and unlabeled data, respectively. **All four proposed methods significantly outperform vanilla DARE (Yu et al., 2023a), demonstrating that (randomized) DPP can still be highly effective, even at extreme pruning rates of 99% parameter reduction, when incorporating our modifications.** Notably, our new post-hoc rescaling achieves a 40-fold model size reduction with minimal impact on performance compared to the unpruned model. See Tables 2,3 for more details.

test performance (see Fig. 1). Thus, these two factors emerge as the main contributors to DARE's failure modes. This analysis inspires two new algorithms that significantly improve DARE's performance, as follows.

• **Drop and rescale with $1/q$ (DAREx-q):** To ensure efficiency at high pruning-rates (e.g., $p > 0.9$), where DPP offers significant savings, our first algorithm DAREx-q modifies DARE's rescaling factor from $1/(1-p)$ to $1/q$, where $q > 1 - p$. While (Yu et al., 2023a) recommended using the factor $1/(1-p)$ to maintain zero-expectation changes in intermediate representations (accounting for DARE's randomness), they neglected the importance of controlling the *absolute* changes and the impact of randomness, particularly since the pruning is applied only once. A large rescaling factor can inflate the variance of these changes; therefore, by tuning the rescaling factor $1/q$, we can effectively balance the mean and variance, thereby minimizing performance degradation. We develop and evaluate four variants of DAREx-q (detailed in Sec. 4.1) to suit different scenarios along two orthogonal axes: (a) whether a *labeled* validation set is available and (b) whether to tune a global $1/q$ across all layers or a per-layer $1/q$. All four methods provide rescaling factors that

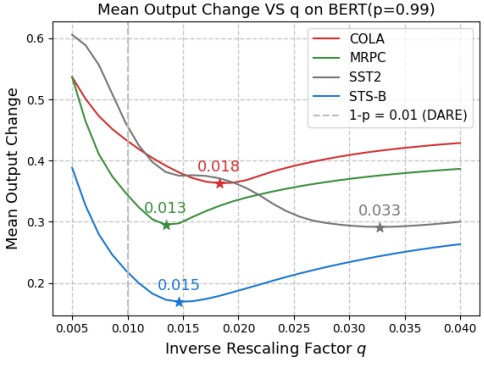
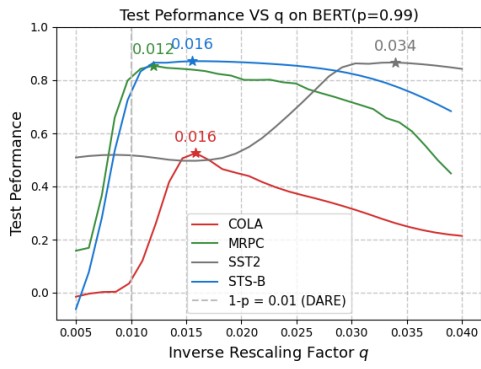

(a) Last-layer output changes on a batch of training data.
(b) Test performance.

Figure 1: Our DAREx-q improves on DARE's performance by tuning the rescaling factor $1/q$. Experiments with BERT at a pruning rate of $p = 0.99$. *Right:* The optimal rescaling factor (asterisk), which maximizes test performance, differs from the standard $1/(1-p)$ across all four datasets and yields up to $> 10$-fold gains. *Left:* **The rescaling factor that minimizes the last-layer output change, averaged over last-layer neurons, serves as an excellent proxy for the optimal factor maximizing performance and can be determined by inference on a single training batch.**

significantly improve upon vanilla DARE while preserving its purely post-training nature, demonstrating that effective DPP remains possible even at high pruning rates.

• **AdamR fine-tuning:** Our second algorithm controls the mean and variance of DPs (averaged over input features at each intermediate layer) prior to applying DARE. Specifically, our AdamR, a modified Adam optimizer, applies $L_2$ regularization directly to the DPs during fine-tuning. Combining AdamR with DARE (DAREx-$L_2$) yields highly competitive performance (see Table 1). While this requires in-training modification, it significantly broadens the scenarios where randomized DPP can be effectively applied.

• **Extensive experiments:** We conduct extensive experiments on both encoder-decoder and decoder-only models across a range of downstream tasks. The results demonstrate the effectiveness of both DAREx-q and DAREx-$L_2$ algorithms. As summarized in Table 1, applying these techniques to a fine-tuned BERT model on the CoLA and SST-2 datasets leads to substantial performance improvements, consistently exceeding 35%. Additional results are presented in Tables 2,3,5,7 and Figures 2,3.

• **Importance-based DPP:** We incorporate importance-based methods such as magnitude-pruning and WANDA into DPP. Our analysis reveals that while random-based DPP generally outperforms important-based DPP, especially under higher pruning rates, combining the latter with AdamR-$L_1$ regularization (as opposed to $L_2$) forms a competitive alternative (see Tables 1 and 7). Moreover, importance-based DPP can outperform random-based DPP when DPs are large and customized fine-tuning is not an option.

## 2 RELATED WORK

**Delta-parameter pruning.** DPs represent the parameter changes induced by fine-tuning and are critical to various operations. In Federated and Distributed Learning, DPs are continuously exchanged between a central server and distributed clients during training (Li et al., 2021; 2020; Khirirat et al., 2018; Chen et al., 2021). In model merging, the goal is to merge multiple task-specific models, fine-tuned from the same pretrained backbone, by combining their DPs into a single model with diverse capabilities (Wortsman et al., 2022; Ilharco et al., 2022). Similarly, in multi-task serving, DPs are processed separately while the base model weights are computed once during the forward pass (Yao & Klimovic, 2023; Liu et al., 2024). (Yu et al., 2023a; Liu et al., 2024; Yao & Klimovic, 2023) focus on reducing the size of DPs, and recently(Yu et al., 2023a) has popularized DARE as a powerful random-based DPP method. However, DARE fails with large

DPs or high pruning rates. We propose techniques that, applied during both fine-tuning and post-hoc stages, restore performance and significantly improve upon vanilla DARE. Our results demonstrate, for the first time, that DPP remains effective even at extreme pruning rates, reducing up to 99% of the model parameters. Thanks to our modification, we also show that random-based DPP can be combined with LoRA and can be used for structured pruning.

**Model parameter pruning.** There are two primary approaches to model parameter pruning: regularization-based and importance-based methods; however, an exhaustive review is beyond the scope of this paper. Among importance-based methods, early works (LeCun et al., 1989; Hassibi & Stork, 1992) introduced importance scores based on second-order information from the loss function. In deep neural networks (DNNs), methods such as (Han et al., 2015a; Li et al., 2018; Han et al., 2015b) utilize magnitude pruning to eliminate weights based on their magnitude. More recently, (Sun et al., 2023) developed an importance score called WANDA, which is proportional to the product of weight magnitude and the norm of the input feature, specifically targeting LLMs. However, the application of these techniques to DPP remains poorly understood. We address this gap by studying both the classical magnitude pruning method and the recently proposed LLM-targeted approach, WANDA, within the context of DPP.

## 3 Delta Parameter Pruning

DPP aims to efficiently store a fine-tuned model $\boldsymbol{\theta}_F$ initialized from a pre-trained base model $\boldsymbol{\theta}_P$ by pruning the difference known as the "delta parameter" (DP), defined as $\boldsymbol{\Delta_\theta} = \boldsymbol{\theta}_F - \boldsymbol{\theta}_P$ between the fine-tuned and the base model. Successful DP pruning (DPP) removes parameters from $\boldsymbol{\Delta_\theta}$ resulting in reduced storage requirements while minimally compromising performance. Sec. 3.1 presents the theoretical model used to analyze various DPP methods. Sec. 3.2 and Appendix A provide a theoretical analysis of randomized and importance-based DPP methods, respectively. These theoretical insights are tested through a detailed controlled experimental analysis using a two-layer neural network in App. B, and, more importantly, applied in Sec. 4 to guide the design of new algorithms that significantly improve LLM performance in Sec. 5.

### 3.1 Formulation

To gain analytical insights, we consider the application of DPP to perceptron operations in a neural network. This focus is partly motivated by the observation that multi-layer perceptron (MLP) operations, both in the projection layers of attention mechanisms and in feed-forward networks, constitute the majority of operations in LLMs. Consider a perceptron layer, where $\boldsymbol{x} \in \mathbb{R}^n$ is the input activation vector connecting to $m$ output neurons, $\boldsymbol{W} \in \mathbb{R}^{m \times n}$ is the hidden-layer weight matrix, and $\sigma$ is a Lipschitz activation non-linearity (e.g., ReLU, sigmoid).[1] Let $\boldsymbol{\Delta_W} \in \mathbb{R}^{m \times n}$ denote the delta parameters of the weight matrix. A pruning mapping $\mathcal{P} : \mathbb{R}^{m \times n} \to \mathbb{R}^{m \times n}$ aims to reduce the parameters in $\boldsymbol{\Delta_W}$, leading to a sparsified $\mathcal{P}(\boldsymbol{\Delta_W})$. The desired sparsity level, $k \in [m]$, is a critical design specification for DPP, often dictated by factors such as storage constraints relative to the number of fine-tuned models. For randomized DPP, elements of $\boldsymbol{\Delta_W}$ are dropped with a probability $p \in [0, 1]$, resulting in an average sparsity level of $k = (1-p)mn$. The objective is to prune the delta parameters to a sparsity level $k$ (or $1 - p$ for randomized DPP) while maintaining the performance of the pruned model, represented by $\boldsymbol{W}_P + \mathcal{P}(\boldsymbol{\Delta_W})$, close to that of the original model $\boldsymbol{W}_P$. To quantify the impact of pruning, we define the difference vector $\boldsymbol{h}^{\mathrm{diff}}$, which measures the change in intermediate output model representations between the pruned and original models:

$$\boldsymbol{h}^{\mathrm{diff}} := \boldsymbol{\Delta_W} \boldsymbol{x} - \mathcal{P}(\boldsymbol{\Delta_W}) \boldsymbol{x}. \tag{1}$$

We require that output after pruning $\sigma\left((\boldsymbol{W}_P + \mathcal{P}(\boldsymbol{\Delta_W}))\boldsymbol{x}\right)$ does *not* differ much from the output before pruning $\sigma\left((\boldsymbol{W}_P + \boldsymbol{\Delta_W})\boldsymbol{x}\right)$. By Lipschitzness of the activation, this translates to each entry $h_i^{\mathrm{diff}}$ of the difference vector $\boldsymbol{h}^{\mathrm{diff}}$ being small in absolute value. Thus, we study how $\mathcal{P}$ affects $|h_i^{\mathrm{diff}}|, i \in [m]$.

---

[1]Bias terms can be included by augmenting the input and DPs to $\boldsymbol{x}' = [\boldsymbol{x}, 1]$ and $\boldsymbol{\Delta'_W} = [\boldsymbol{\Delta_W}, \boldsymbol{\Delta_b}]$, respectively, but recent LLM architectures often omit bias terms for training stability (Touvron et al., 2023; Chowdhery et al., 2023)

## 3.2 RANDOMIZED DPP: RANDOM DROP AND RESCALE (DARE)

DARE randomly sets each element of $[\mathcal{P}(\boldsymbol{\Delta_W})]_{ij}$ to zero (dropped) with probability $p$, and *rescales* non-zero elements to $[\mathcal{P}(\boldsymbol{\Delta_W})]_{ij} = [\boldsymbol{\Delta_W}]_{ij}/(1-p)$(Yu et al., 2023a). While DARE has been tested on a wide range of fine-tuned models across various tasks, it exhibits performance degradation under high pruning rates, as shown in Table 7 and Table 8. We seek the underlying causes of these limitations and explore potential improvements. Our approach involves quantifying the difference vector $\boldsymbol{h}^{\mathrm{diff}}$ in relation to key variables: the dropout rate, the DPs, and the input data. As per Eq. 1, the effect of DARE on the output becomes

$$h_i^{\mathrm{diff}} = \sum_{j\in[n]} \Delta W_{ij}x_j - \sum_{j\in[n]} \frac{1}{1-p}\delta_{ij}\Delta W_{ij}x_j, \qquad (2)$$

where $p$ is the *drop rate*, $\{\delta_{ij}\}_{i\in[m],j\in[n]}$ are iid Bernoulli$(1-p)$ random variables, and $1/(1-p)$ is the *rescaling factor*. We denote $\Delta W_{ij}$ the $(i,j)$ entry of $\boldsymbol{\Delta_W} \in \mathbb{R}^{m\times n}$.

It is easy to see that $\mathbb{E}[h_i^{\mathrm{diff}}] = 0$ (expectation over $\delta_{ij}$), and Yu et al. (2023a) use this as justification for DARE's effective performance. However, this alone overlooks the sources of DARE's failure modes, which we reveal by instead establishing bounds on the *absolute* output changes $|h_i^{\mathrm{diff}}|$. The following theorem addresses this by providing a high-probability bound on these changes, with respect to DARE's randomness, which arises from the Bernoulli variables. See App. E.1 for the proof.

**Theorem 3.1.** *Denote $h_i^{\mathrm{diff}}$ as the $i$-th component of $\boldsymbol{h}^{\mathrm{diff}}$ in Eq. (2). For $i \in [m], j \in [n]$, let $c_{ij} = \Delta W_{ij}x_j$ represent the change in influence of the $j$-th feature on the $i$-th output neuron after fine-tuning. Define[2] the mean $\bar{c}_i$ and variance $\sigma_i^2$ of these as: $\bar{c}_i = (1/n)\sum_{j\in[n]} c_{ij}$ and $\sigma_i^2 = (1/n)\sum_{j\in[n]}(c_{ij} - \bar{c}_i)^2$. Then, for any $i \in [m]$ and $\gamma \in (0,1)$, it holds with probability at least $1 - \gamma$ that*

$$|h_i^{\mathrm{diff}}| \le (\Psi(p)/(1-p))\sqrt{n(\bar{c}_i^2 + \sigma_i^2)}\sqrt{\log(2/\gamma)} \,,$$

*where $\Psi(p) = (1-2p)/\log((1-p)/p)$ if $p \le 1/2$, otherwise $\Psi(p) = \sqrt{2p(1-p)}$.*

Thus, the key factors influencing the magnitude of $|h_i^{\mathrm{diff}}|$ are: (i) the rescaling factor $1/(1-p)$ and (ii) the mean and variance (averages are over input dimension $j \in [n]$) of the influence parameters $\{c_{ij}\}$. Specifically, increasing the rescaling factor (eqv. increase drop-rate $p$) increases $|h_i^{\mathrm{diff}}|$ at most a rate of $\mathcal{O}((1-p)^{-\frac{1}{2}})$, which can be large in demanding pruning scenarios when $p$ is large. Also DARE yields smaller $|h_i^{\mathrm{diff}}|$ values when $c_{ij} = \Delta W_{ij}x_j$ exhibit low first (mean) and second order (variance) averages. We validate these conclusions through elaborate experiments on a controlled setup in Appendix B. More importantly, our algorithms in Sec. 5, further tested on LLMs, are also built upon the insights from this theorem and its observations.

## 3.3 IMPORTANCE-BASED DPP

Using the same analytical framework, we extend the application of importance-based pruning methods, specifically magnitude pruning and WANDA, to the DPP setting. Due to space constraints, we provide a detailed discussion in App. A. In brief, importance-based DPP can achieve strong performance when the distribution of the coefficients $c_{ij}$ exhibits light tails and sharp-peakedness. We provide empirical validation of these findings in Sec. C.5 and Appendix B.

## 4 ALGORITHMS

To extend the applicability of DPP at high pruning rates and in cases where DPs of fine-tuned models exhibit undesirable statistics, we propose here two strategies based on the theoretical insights discussed in Sec. 3.

---

[2]We call 'mean'/'variance' here the first/second -order summary statistics of $\{c_{ij}\}_{j\in[n]}$ over the input dimension. On the other hand, the theorem's 'high-probability' statement is w.r.t. the randomness of random variables $\{\delta_{ij}\}_{j\in[n]}$

## 4.1 ADJUSTING THE RESCALING FACTOR

**Motivation.** Recall from Thm. 3.1 as $p$ increases, the absolute difference in outputs $|h_i^{\text{diff}}|$ grows at at most[3] a rate of $\mathcal{O}((1-p)^{-\frac{1}{2}})$. Setting the rescaling factor to $q = 1 - p$ eliminates the mean of $h_i^{\text{diff}}$, but does not minimize its magnitude. Through empirical analysis, we demonstrate in Fig. 1 how various values of $q$ influence model outputs in terms of both mean output change (i.e., the average of $|h_i^{\text{diff}}|$ across all last-layer neurons) in Fig. 1a, and, test performance in Fig. 1b across different datasets. For 99% pruning rate $(1 - p = 0.01)$ we observe these quantities over a range of $q$ values from 0.005 to 0.04. The results depicted in Fig. 1a for mean output change, indicate a convex-like trend, with minimum points (marked with asterisks) achieved at $q$ values higher than the vanilla $1 - p$. E.g., the optimal $q$ for COLA and SST2 is $\approx 0.018$ and 0.033, respectively. Recall now that in our analysis, the amount of change in output activations is used as a proxy for test performance: smaller mean output change corresponds to better performance. Fig. 1b validates this: the test-performance curves show a concave-like shape, with the peak performance also occurring at $q > 1 - p$ and numerically consistent with the values of that minimize the mean absolute output change.

**Drop and rescale with $1/q$ (DAREx-q).** Motivated by these observations, we propose two variants of DAREx-q: tuning based on (1) maximizing performance on a labeled validation set, and (2) minimizing mean output change over unlabeled data.
• *DAREx-q with labeled validation data* $(1/q_v)$: We use a validation dataset $\{x_v, y_v\} \in \mathcal{V}$ to determine the best rescaling factor $1/q_v$ that maximizes test performance (eqv. minimizes test error) on the validation set. Specifically, we select $q_v = \arg\min_q \mathbb{P}_{\mathcal{V}}(f_q(x_v) \neq y_v)$, where $f_q$ represents the pruned model rescaled by $1/q$. We then randomly drop a faction $p$ of the model DPs and rescale those that are not dropped with $1/q_v$. Further details can be found in Algorithm 2 (1) in the Appendix.
• *DAREx-q with unlabeled data* $(1/q_e)$: This method selects $q$ by optimizing an unsupervised objective measuring the mean output difference $|h_i^{\text{diff}}|$ across all last-layer neurons of the model. This approach is based on the observation in Fig. 1 that mean output difference is as an effective unsupervised proxy for test performance. The key advantage of this method over $q_v$ is that it does *not* require labeled data. Specifically, in our implementations, selecting $q_e$ involves running inference on a single fixed batch of data (with texts from either the training data or any other natural dataset) using both the pruned and original models. Refer to Algorithm 2 in the Appendix for further details.

**DAREx-q with per-layer tuning.** The implementations of DAREx-q described above select a single, global rescaling factor $(1/q_v$ or $1/q_e)$ that is then applied to all surviving DPs across all layers of the network. However, since the DPs in intermediate layers each have their own mean output change over their respective output layers, a natural question arises: Can we select a different rescaling factor for each layer $\ell \in [L]$? Moreover, can this be done efficiently without requiring $L$ hyperparameter searches (one for each layer), which would be computationally prohibitive? In Appendix C.9, we demonstrate that the answer to both of these questions is affirmative. To circumvent the computational challenge of $L$ searches, we leverage theoretical insights by extending Thm. 3.1 to provide a high-probability bound for arbitrary $q$ and then optimizing over it. Like Thm. 3.1, the new bound still depends on the first- and second-order statistics of the coefficients $c_{ij}^{\ell}$, which means the optimal $q^{\ell}$ can vary per layer $\ell$ based on these statistics. We denote this vector of per-layer rescaling factors as $1/\mathbf{q}_v = [1/q_v^1, \ldots, 1/q_v^L]$ and $1/\mathbf{q}_e$, respectively, for two implementations depending on whether the performance on validation set or mean output change is optimized. Please refer to App. C.9 for details. In Sec. 5.1, we show that this approach performs very well on encoder models and is comparable to the global $1/q$ approach for decoder models.

## 4.2 ADAMR FINETUNING

**Motivation.** While DAREx-q addresses the issue of large rescaling factors at high pruning rates, we've seen that for absolute output changes $|h_i^{\text{diff}}|$ to remain small, the first- and second-order averages of the influence

---

[3]The rate of growth with $p$ in the high-probability upper bound of Thm. 3.1 is tight and thus predictive of the output change taking larger values as $p$ increases; see App. E.1.

Table 2: **DAREx-q consistently outperforms DARE across four datasets and two encoder models.** The pruning rate is set to $p = 0.99$. **Bold** indicates the best performance, while underline marks the second-best.

| Dataset | SST2 | | COLA | | MRPC | | STS-B | |
|---|---|---|---|---|---|---|---|---|
| Models | BERT | RoBERTa | BERT | RoBERTa | BERT | RoBERTa | BERT | RoBERTa |
| No Pruning | 90.25 $(-)$ | 92.09 $(-)$ | 56.24 $(-)$ | 62.10 $(-)$ | 85.01 $(-)$ | 90.05 $(-)$ | 88.51 $(-)$ | 90.57 $(-)$ |
| DARE | 51.00 (0.64) | 51.81 (1.00) | 4.25 (2.83) | 7.65 (3.63) | 79.96 (6.07) | 87.32 (0.69) | 82.56 (1.13) | 81.59 (1.44) |
| DAREx-q $(1/q_v)$ | 85.64 (3.07) | 86.56 (4.59) | 48.96 (4.13) | 53.58 (1.97) | **83.92 (0.62)** | 87.30 (0.58) | 86.60 (1.61) | 88.07 (0.70) |
| DAREx-q $(1/\mathbf{q}_v)$ | **89.60 (0.67)** | **90.40 (1.36)** | **53.12 (2.01)** | **58.38 (2.05)** | 83.86 (0.66) | **89.80 (0.60)** | 87.55 (0.12) | **89.10 (0.32)** |
| DAREx-q $(1/q_e)$ | 85.81 (2.72) | 81.05 (4.96) | 45.20 (2.04) | 52.10 (5.01) | 83.12 (1.11) | 88.85 (0.26) | 87.03 (0.12) | 87.29 (0.57) |
| DAREx-q $(1/\mathbf{q}_e)$ | 87.39 (0.86) | 89.62 (1.35) | 51.70 (2.57) | 54.43 (2.49) | 83.25 (0.94) | 89.59 (0.42) | **87.59 (0.19)** | 87.60 (0.49) |

Table 3: Similar to Table 2, **DAREx-q yields significant gains for decoder models**.

| Dataset | GSM8K | | | | | | | |
|---|---|---|---|---|---|---|---|---|
| Methods | Abel-7B | | MetaMath-7B | | WizardMath-7B | | MetaMath-Qwen2-0.5B | |
| | $p = 0.95$ | $p = 0.99$ | $p = 0.95$ | $p = 0.99$ | $p = 0.95$ | $p = 0.99$ | $p = 0.95$ | $p = 0.99$ |
| No Pruning | 58.32 | | 65.50 | | 54.90 | | 42.00 | |
| DARE | 37.30 | 0.00 | 58.22 | 0.00 | 47.10 | 0.00 | 0.00 | 0.00 |
| DAREx-q $(1/q_v)$ | **47.20** | 20.20 | 59.05 | **34.87** | 49.81 | **35.63** | **30.70** | **19.17** |
| DAREx-q $(1/\mathbf{q}_v)$ | 44.50 | 20.00 | 59.28 | 32.06 | **50.64** | 34.34 | 29.56 | 18.65 |
| DAREx-q $(1/q_e)$ | 42.99 | **21.30** | **60.34** | 32.60 | 49.05 | 33.97 | 30.32 | 18.95 |
| DAREx-q $(1/\mathbf{q}_e)$ | 42.84 | 19.63 | 59.28 | 28.05 | **50.64** | 33.58 | 28.80 | 17.66 |

factors $\{c_{ij}\}_{j \in [n]}$ must also be small. These factors directly depend on the magnitude of the DPs $\Delta W_{ij}$, which are determined during fine-tuning. We propose an in-training method that fine-tunes the model to keep these statistics small, thereby guaranteeing that the post-hoc application of DARE maintains performance.

**AdamR-$L_2$.** Our approach is to fine-tune the model with $L_2$ regularization on the DPs $\mathbf{\Delta_\theta} = \mathbf{\theta}_F - \mathbf{\theta}_P$. As mentioned, the motivation behind this is penalizing $\|\mathbf{\Delta_\theta}\|_2$ directly translates to smaller total second-order averages $\sum_{j \in [n]} c_{ij}^2$, which, as previously noted, capture the extent of change in the output layer post-finetuning and factor into the bound in Thm. 3.1. To implement this, we replace the weight decay of AdamW—the standard choice for training LLMs—with our custom regularization. Specifically, we adjust the regularization strength based on the gradient norm, as Xie et al. (2024) recently found that weight decay can cause large gradient norms during the final phase of training. Our modified decay step of AdamW is:

$$\mathbf{\theta}_t = \mathbf{\theta}_{t-1} - \frac{\eta}{\sqrt{\hat{\mathbf{v}}_t} + \epsilon} \hat{\mathbf{m}}_t - \frac{\eta}{\sqrt{\bar{v}_t} + \epsilon} \lambda(\mathbf{\theta}_{t-1} - \mathbf{\theta}_P), \tag{3}$$

where $\mathbf{\theta}_t$ represents model parameters at iteration $t$, $\hat{\mathbf{m}}_t$ and $\hat{\mathbf{v}}_t$ are the first and second moments of gradients, and $\bar{v}_t$ is the mean of the second moment used to adjust regularization. See also Algorithm 1 in Appendix.

**AdamR-$L_1$.** We extend our approach to fine-tune the model in preparation for importance-based pruning, rather than random-based pruning, post-training. In this case, we regularize using the $L_1$ norm of the DPs. This is again motivated by the analysis in Sec. 3.3. The detailed algorithm, referred to as AdamR-$L_1$, is provided in Algorithm 1 in the Appendix.

## 5 EXPERIMENTS

We validate the effectiveness of our analysis and proposed algorithms through comprehensive experiments. **Datasets and models for Encoder-based and Decoder-based LMs.** For Encoder-based LMs, we utilize four datasets—sentence acceptability dataset CoLA (Warstadt et al., 2019), sentiment detection dataset SST-2 (Socher et al., 2013), paraphrase dataset MRPC (Dolan & Brockett, 2005), and sentence similarity dataset STS-B (Cer et al., 2017). For the selection of pretrained backbones, we choose BERT-base-uncased (Devlin et al., 2018) and RoBERTa-base (Liu et al., 2019), fine-tuning them on these task-specific datasets to obtain supervised finetuned models. For Decoder-based LMs, we focus on mathematical reasoning tasks. Due to

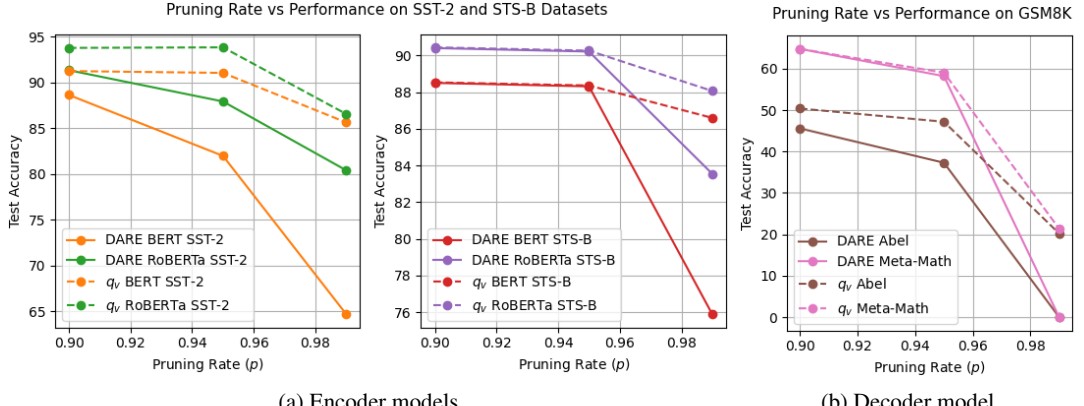

(a) Encoder models                                                          (b) Decoder model

Figure 2: **Across pruning rates** $p$**, DAREx-q performs at least as well as vanilla DARE and significantly outperforms it at higher pruning rates.**

constraints on resources for fine-tuning, we opted to fine-tune a smaller Qwen2-0.5B (Yang et al., 2024) model on the MetaMath dataset (Yu et al., 2023b). Additionally, we utilize publicly available mathematical reasoning models, including the MetaMath-llema-7B (Yu et al., 2023b), MetaMath-7B (Yu et al., 2023b), WizardMath-7B (Luo et al., 2023) and Abel-7B (Chern et al., 2023), all based on the Llama2-7B architecture (Touvron et al., 2023). We then use the GSM8K (Cobbe et al., 2021) to test these models.

**Evaluation metrics.** Following (Yu et al., 2023a), we evaluate performance using various metrics: the Matthews correlation coefficient for CoLA, accuracy for SST-2, a combined score of accuracy and F1 for MRPC, the mean of Pearson and Spearman correlations for STS-B, and zero-shot accuracy for GSM8K.

## 5.1 RESCALING-PARAMETER MODIFICATION

We evaluate our DARq algorithm, as discussed in Section 4.1. Recall the four proposed variations for choosing the rescaling factor, categorized by two distinct factors: (a) the use of labeled validation data or not ($1/q_v$ or $1/q_e$, respectively); and (b) global tuning versus per-layer tuning ($1/q$ versus $1/\mathbf{q}$).

**Encoder models.** Table 2 compares DAREx-q (all four variants) with vanilla DARE (with rescaling factor $1/(1-p)$). We report average performance and standard deviation on the test set over four independent runs. For reference, we also report performance of the finetuned model before pruning. We find that: (1) *All* DAREx-q variants outperform vanilla DARE in *all* cases and by a large margin (e.g., $> 40\%$ on COLA). (2) $1/q_e$ and $1/\mathbf{q}_e$ deliver competitive performance compared to $1/q_v$ and $1/\mathbf{q}_v$, making it an alternative when a labeled validation set is unavailable. (3) Per-layer rescaling ($1/\mathbf{q}_v$ and $1/\mathbf{q}_e$) outperforms global rescaling.

**Decoder models.** Table 3 evaluates DAREx-q on decoder models. We find that: (1) DAREx-q consistently boosts performance over vanilla DARE across *all* models and both pruning rates $p = 0.95, 0.99$. (2) Higher pruning rates yield striking improvements: vanilla rescaling fails completely, while DAREx-q yields non-trivial performance; (3) Global rescaling ($1/q$) provides performance comparable to that of per-layer rescaling ($1/\mathbf{q}$), making it a more efficient recommendation for decoder models.

**Performance across different pruning rates.** Fig. 2 compares performance of DAREx-q to vanilla DARE for varying values of pruning rates $p$ for both encoder and decoder models. For *all* examined values of $p$, DAREx-q is at least as good as vanilla DARE. More importantly, we confirm that DAREx-q significantly outperforms vanilla DARE at higher pruning rates.

Table 4: **DAREx-q is orthogonal to and can be combined with LoRA**, leading to significant improvements over vanilla DARE.

| Datasets ($p = 0.9$) | STS-B | COLA | SST2 | MRPC |
|---|---|---|---|---|
| No Pruning | 90.07 | 59.56 | 94.27 | 87.09 |
| DARE (Yu et al., 2023a) | 81.49 | 50.96 | 48.05 | 80.03 |
| DAREx-q ($1/q_v$) (ours) | **85.06** | **56.73** | **90.25** | **82.53** |
| DAREx-q ($1/q_e$) (ours) | 85.00 | **56.73** | 90.02 | **82.53** |

Table 5: **DARE with AdamR-$L_2$ finetuning (DAREx-$L_2$) significantly improves performance on encoder models compared to DARE without it.**

| Dataset | p | BERT | | | RoBERTa | | |
|---|---|---|---|---|---|---|---|
| | | original | $L_2$ | difference | original | $L_2$ | difference |
| MRPC | No Pruning | 85.01 (-) | 84.31 (-) | -0.70 | 90.05 (-) | **90.69** (-) | +0.64 |
| | 0.99 | 79.96 (6.07) | **84.30 (0.48)** | **+4.34** | 87.32 (0.69) | **90.02 (0.41)** | **+2.70** |
| STS-B | No Pruning | 88.51 (-) | 87.99 (-) | -0.52 | 90.57 (-) | 90.25 (-) | -0.32 |
| | 0.99 | 82.56 (1.13) | **87.62 (0.27)** | **+5.06** | 81.59 (1.44) | **89.88 (0.29)** | **+8.29** |
| SST2 | No Pruning | 90.25 (-) | 89.91 (-) | -0.34 | 92.09 (-) | 92.89 (-) | +0.80 |
| | 0.99 | 51.00 (0.64) | **88.17 (0.44)** | **+37.17** | 51.81 (1.00) | **91.95 (0.24)** | **+40.14** |
| COLA | No Pruning | 56.24 (-) | **59.27** (-) | +3.03 | 62.10 (-) | 59.27 (-) | -2.83 |
| | 0.99 | 4.25 (2.83) | **57.24 (0.62)** | **+52.99** | 7.65 (3.63) | **59.01 (1.26)** | **+51.36** |

**LoRA+DAREx-q:** As a post-training DPP, DAREx-q is orthogonal to and can be combined with parameter-efficient tuning approaches like LoRA (Hu et al., 2021). To demonstrate this, we fine-tune the BERT-Base model by training only the LoRA module, then prune the parameters of LoRA at rate $p = 0.9$. The results, shown in Table 4, reveal that applying vanilla DARE does *not* yield good performance. In contrast, our DAREx-q significantly improves the results. In addition to these, we demonstrate in Appendix Sec. C.11 that DAREx-q is even comparable to sparse fine-tuning, despite being a purely post-training method.

**Structural DPP with DAREx-q.** Our random-based DAREx-q can also be utilized for structural DPP for enhanced hardware efficiency (Shen et al., 2022; Yao & Klimovic, 2023). In this setup, we randomly select $a\%$ of the input dimensions of a MLP layer and randomly retain $b\%$ of the DPs within the selected dimensions, achieving a total structured DP pruning rate of $p = a \cdot b\%$. Table 9 in the Appendix, uses $a = 5, b = 20$ for an equivalent $p = 0.99$, to show that DAREx-q outperforms DARE with a 40% improvement.

### 5.2 ADAMR-$L_2$ FINETUNING

We now show that our proposed in-training method of Sec. 4.2 successfully achieves highly prunable DPs.

**Encoder models.** Table 5 compares the performance of DARE applied to the original finetuned model against its performance when applied to models finetuned with AdamR-$L_2$ across two encoder models and four datasets. We set the pruning rate at $p = 0.99$ and, for reference, also report the performance of the finetuned models without pruning, both with and without AdamR-$L_2$. Consistent with the insights of Sec. 4.2, applying DARE after AdamR-$L_2$ (DAREx-$L_2$) outperforms vanilla DARE without regularization. Additionally, the performance is only marginally lower than that of the original finetuned model. Figure 7 in the Appendix further shows that increasing the regularization strength allows pruning up to 99.9% of parameters with only a minor performance drop. See Appendix C.4 for further ablations on regularization weight. Finally, Table 9 in the Appendix demonstrates that DAREx-$L_2$ unlocks the potential of random-based DPP for structural pruning, improving DARE without $L_2$ finetuning by over 40%. We also conduct training analysis of DAREx-$L_2$ in Appendix F.

**Decoder models.** To demonstrate the effectiveness of DAREx-$L_2$ on decoder-based LMs, we finetune Qwen-0.5B using the

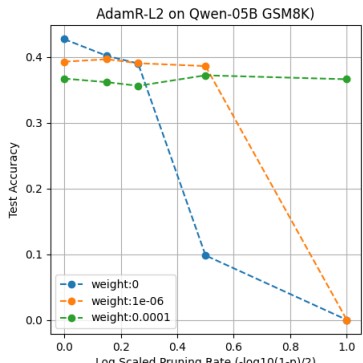

Figure 3: Applying DARE on decoder models finetuned with AdamR-$L_2$ (DAREx-$L_2$) at varying regularization strengths demonstrates significant performance improvements for $p \geq 0.9$.

MetaMath dataset (Yu et al., 2023b). Figure 3 shows the $L_2$ fine-tuned Qwen-0.5B model with different regularization weights on the MetaMath datasets. We used pruning rates $p \in [0, 0.5, 0.7, 0.9, 0.99]$ to prune the DPs. For visualization, the x-axis uses a log-scaled pruning value of $-\log(1-p)/2 = [0, 0.15, 0.26, 0.5, 1.0]$. When AdamR-$L_2$ is not applied (regularization weight $= 0$), performance on GSM8K drops significantly at

$p = 0.9$. By increasing the regularization weight, higher pruning rates can be achieved. With a regularization weight of $1e^{-4}$ (green line), we can reach a $99\%$ pruning rate while maintaining performance.

### 5.3 HOW TO APPLY IMPORTANCE-BASED DPP AND WHEN IS IT COMPETITIVE?

We have seen, both theoretically and empirically, that DARE is effective at pruning DPs when their first and second order statistics (e.g. as summarized by $\bar{c}_i$ and $\sigma_i^2$ in Thm. 3.1) are small. While most fine-tuned LLMs typically introduce small DPs Yao & Klimovic (2023); Lee et al. (2019), exceptions exist. For instance, Table 8 in the App. shows that the MetaMath-LLaMA-7B model, fine-tuned on large datasets (MetaMathQA Yu et al. (2023b) and Proof-Pile-2 Azerbayev et al. (2023)), has a mean delta parameter magnitude of 0.017, notably larger than that of Abel-7B and MetaMath-7B. In such cases, as demonstrated in Table 6, DARE experiences a substantial performance drop when the pruning rate exceeds 0.1.

Can importance-based DPP methods serve as alternatives? Yes, as shown in Table 6. Importance-based methods like MP and WANDA can maintain pruning rates of up to 0.5 with only minimal performance degradation. Thus, these methods are effective alternatives to DARE when DPs are large and fine-tuning with customized methods like AdamR-$L_2$ is not feasible.

Table 6: Peformance of pruning methods on MetaMath-llema.

| Methods | GSM8K | | | | |
|---|---|---|---|---|---|
| (MetaMath-llema-7B) | $p = 0.0$ | $p = 0.1$ | $p = 0.3$ | $p = 0.5$ | $p = 0.6$ |
| MP | | **63.00** | 61.00 | 47.50 | 32.00 |
| WANDA | 64.50 | **63.00** | **62.50** | **54.50** | **32.50** |
| DARE | | 51.00 | 0.00 | 0.00 | 0.00 |
| Random Drop | | 52.50 | 6.50 | 0.00 | 0.00 |

**Improvements with AdamR-$L_1$.** For completeness of our study, we also demonstrate that finetuning with AdamR-$L_1$ can boost performance of importance-based DPP as suggested in Sec. 4.2. Specifically, in Table 7 in App. C.5, we show elaborate experiments comparing the performance of MP and WANDA (applied to DPs) when used on a model finetuned with/without AdamR-$L_1$ for varying values of pruning rates and two encoder models across four datasets. In summary, we find: (1) AdamR-$L_1$ consistently increases performance of MP. (2) While AdamR-$L_1$ improves WANDA's performance for most datasets, this is not always the case. We attribute this to WANDA's importance score being impacted not only by DPs $\Delta W_{ij}$ but also by input magnitude $|x_j|$ and important DPs not always being aligned with outlier activations (see App. D).

## 6 CONCLUSIONS AND LIMITATIONS

Our study of DPP methods, provides theoretical insights, extensive experiments in controlled environments, and demonstrates practical applications on large-scale models and datasets. Starting with an in-depth analysis of random-based DPP methods, we identify key factors that degrade their performance. Based on these insights, we propose: (i) DAREx-q, a post-training rescaling modification strategy that significantly outperforms the state-of-the-art at high pruning rates and performs at least as well across other rates; and (ii) AdamR finetuning to enhance existing models that struggle with DPP, ensuring the production of highly prunable DPs. Our results, including dramatic gains over baseline methods shown in Table 1, suggest that these strategies warrant further investigation across different tasks and modalities. Additionally, preliminary results suggest that DAREx-q can be

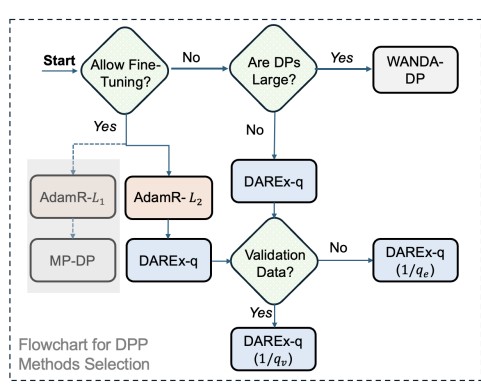

Figure 4: Flowchart for selecting appropriate DPP methods based on different scenarios.

effectively combined with parameter-efficient methods like LoRA and utilized for structural DPP, both of which highlight the potential for further study. Finally, while finetuning LLMs typically results in small DPs favoring random-based DPP, we demonstrate that importance-based DPP methods can serve as robust alternatives for larger DPs. This highlights the need for further exploration of alternatives. In conclusion, our comprehensive study of DPP methods provides a framework for selecting appropriate DPP methods under different scenarios, which we summarize in Fig. 4 (details in App. C.3). Broader impact see App. C.13.

**Acknowledgments:** This work was partially funded by an Alliance Mission Grant, the NSERC Discovery Grant No. 2021-03677, the Alliance Grant ALLRP 581098-22, NFRFE-2023-00936, Natural Sciences and Engineering Research Council of Canada (NSERC), Canada CIFAR AI Chairs program, Canada Research Chair program, MITACS-CIFAR Catalyst Grant Program, the Digital Research Alliance of Canada, Institute of Information & communications Technology Planning & Evaluation (IITP) grant funded by the Korea government (MSIT).

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

# Contents

## A    IMPORTANCE-BASED DPP

In this section, we adapt importance-based pruning, which is traditionally popular for model parameter pruning, e.g. Han et al. (2015b); Li et al. (2018); Sun et al. (2023), into DPP.

**Magnitude pruning (MP).** Magnitude pruning Li et al. (2018) drops model weights according to their magnitudes: less "important" weights with smaller magnitudes are dropped. To extend its application to pruning delta parameters we evaluate importance in terms of the magnitudes of delta parameters, represented as $\Delta S_{ij} := |\Delta W_{ij}|$.

**Pruning based on both weights and activations (WANDA).** Recently, WANDA was introduced by Sun et al. (2023) and was empirically found to achieve state-of-the-art pruning performance for LLMs. It evaluates the importance of each model weight using a score metric that scales the magnitude of the weight by the Euclidean norm of the input feature, thus accounting for the input activations. We extend WANDA to encompass delta parameters by adjusting the importance score to $\Delta S_{ij} = |\Delta W_{ij}| \cdot \|\boldsymbol{x}_j\|_2$.

For their respective scores, MP and WANDA (and any other importance-sampling method adapted to DPP) maintain the top-$k$ parameters in $\boldsymbol{\Delta_W}$ with the highest scores. Let $\mathcal{S}_k \subset [m] \times [n]$ be the set of indices $(i, j)$ that correspond to weights with the top-$k$ scores $\Delta S_{ij}$. Moreover, let $k := (1 - p)mn$ so that the method retains the same number of delta parameters as DARE does in expectation. Accordingly, denote the set of selected weights with respect to $p$ as $\mathcal{S}_p := \mathcal{S}_k$. Then, putting in Eq. (1), the change for each output activation $i \in [m]$ can be expressed as:

$$h_i^{\mathrm{diff}} = \sum\nolimits_{\{j:(i,j)\notin \mathcal{S}_p\}} \Delta W_{ij} x_j \,, \tag{4}$$

where the summation extends over all input dimensions $j$ for which the $(i, j)$ entry of $\boldsymbol{\Delta_W}$ is dropped due to a low score. An importance sampling method can perform effectively even for large $p \approx 1$ if the cumulative contributions from the summation are approximately zero out. This is guaranteed when the distribution of the entries $[\Delta W_{ij} x_i]_{j \in [n]}$ has a light tail and high peakedness. We validate this in Secs. C.5 and B and accordingly propose AdamR-$L_1$ fine-tuning to enhance their pruning performance (algorithm detailed in in Sec. C.7)

## B    ANALYSIS ON TWO-LAYER NEURAL-NETWORK

Having gained analytical insights into the key factors that influence the performance of DPP methods in Sec. 3.2 and Sec. A, we now explore in detail how these factors influence DPP in a two-layer neural network. Concretely, for an input $\boldsymbol{x} \in \mathbb{R}^n$, the model output is $f(\boldsymbol{x}) = \boldsymbol{W}_o \, \mathrm{N}(\phi(\boldsymbol{W}_1 \, \mathrm{N}(\boldsymbol{x}) + \boldsymbol{b}_1)) + \boldsymbol{b}_o$. Here, N denotes layer normalization (RMSnorm Zhang & Sennrich (2019) in our case), $\phi$ is the ReLU activation function, $\boldsymbol{W}_o$ / $\boldsymbol{b}_o$ and $\boldsymbol{W}_1$ / $\boldsymbol{b}_1$ are the weights / biases of the output and the hidden layer respectively, and are all trainable (during both pre-training and fine-tuning). In our experiments, we pre-train the model on the CIFAR-10 dataset and use the SVNH dataset Sermanet et al. (2012) for the supervised fine-tuning task.

**Influence of variance and mean statistics on DARE (Fig. 5a):** Thm. 3.1 establishes how DARE's performance, approximated by the magnitude of $h_i^{\mathrm{diff}}$, is influenced by the mean and variance statistics of $\{c_{ij}\}$. Specifically, smaller values are favorable for performance. To verify this, we compare performance of DARE when the model is fine-tuned using $L_1/L_2$ regularization with respect to the pre-trained parameters. Formally, recalling that $\boldsymbol{\Delta_\theta} = \boldsymbol{\theta}_F - \boldsymbol{\theta}_P$ denotes the delta parameter of all trainable weights, we use regularization that penalizes the following: $\|\boldsymbol{\Delta_\theta}\|_r$, $r \in \{1, 2\}$. $L_2$ regularization directly corresponds to a smaller total energy ($\sum_{ij} c_{ij}^2$) of the $\{c_{ij}\}$ parameters, which recall capture the degree of change in the output layer after fine-tuning. This enters the bound in Thm. 3.1 and suggests that $L_2$ regularization improves DARE's performance. On the other hand, $L_1$ regularization favors sparse $\boldsymbol{\Delta_\theta}$, thus also $\{c_{ij}\}$. Intuitively,

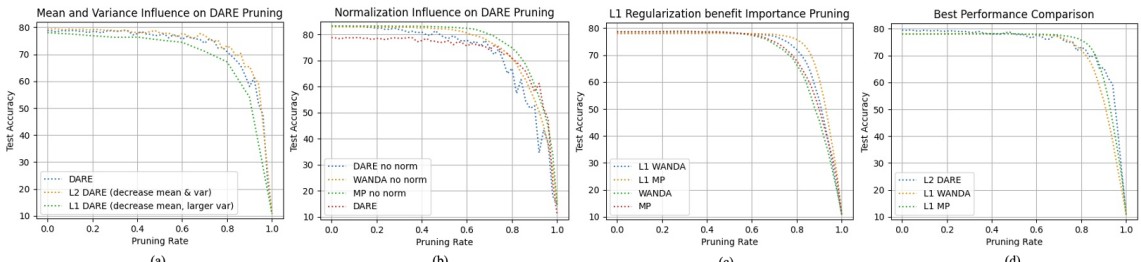

Figure 5: Controlled experiments of DPP performance on two-layer neural net. (a) Influence of variance and mean statistics on DARE. (b) Influence of normalization layer. (c) $L_1$ regularization for importance based pruning. (d) Methods with best-fitting regularization.

this increases the variance, thus we expect from Thm. 3.1 that $L_1$ regularization might hurt performance. Fig. 5 (a) shows the test accuracy of DARE for varying values of pruning rate $p$ for three finetuning settings: no regularization (blue), $L_2$ regularization (orange), and $L_1$ regularization. The results confirm the insights above and thus support Thm. 3.1.

**Influence of normalization layer (Fig. 5b):** Normalization layers have become standard practice in deep learning due to their role in stabilizing training by normalizing the variance of activations, as established in Xu et al. (2019). In this study, we explore the impact of these layers on the performance of DARE. According to Thm. 3.1, reduced variance in activations (denoted as $x_j$s in our theorem setup) leads to decreased variance in the $\{c_{ij}\}$ coefficients, which in turn is predicted to enhance DARE's effectiveness. This hypothesis is confirmed in Fig. 5 (b), where the absence of a normalization layer is shown to expedite the performance decline of DARE as pruning rates increase. Conversely, importance-based methods discussed in Sec. A demonstrate a higher tolerance to increased pruning rates without normalization. These observations are consistent with the discussions in Sec. A.

$L_1$ **regularization for importance based pruning (Figure 5c):** In Sec. A, we discussed how importance-based DPP methods are favored. Here, we demonstrate this is indeed the case by using $L_1$ regularization during finetuning to induce sparse delta parameters, which increases the variance. Concretely, as illustrated in Fig. 5c, introducing $L_1$ regularization improves the performance of importance-based methods. Note also that, in this scenario, magnitude pruning outperforms WANDA due to the absence of outlier features Wei et al. (2022); Sun et al. (2023).

**Methods with best-fitting regularization(Fig. 5d):** In Fig 5d, we compare three methods with their corresponding best-fit regularization. As discussed above, DARE benefits from $L_2$ regularization, while importance-based methods benefit from $L_1$ regularization. We find that that DARE with $L_2$ regularization is more robust with respect to the increase in pruning rate. On the other hand, importance based methods with $L_1$ regularization perform the best for medium-range pruning rates.

## C  ADDITIONAL DETAILS

### C.1  ADDITIONAL RELATED WORK

**Supervised fine-tuning of language models.** Supervised fine-tuning (SFT) of pre-trained LLMs is designed to enhance their capabilities by training on task-specific data, establishing a standard in natural language processing Dodge et al. (2020); Zhao et al. (2023). SFT can be categorized into full fine-tuning Devlin et al. (2018); Liu et al. (2019) and parameter-efficient fine-tuning (PEFT) Ding et al. (2023); Li & Liang (2021); Hu et al. (2021); Deng et al. (2024). However, recent studiesChen et al. (2022); Lu et al. (2023);

Yao & Klimovic (2023) suggest that PEFT methods may not yet achieve the model quality of full parameter fine-tuning, particularly in high-resource tasks Chen et al. (2022). Additionally, Liu et al. (2024) indicates that most models on the Open LLM Leaderboard are derived from full parameter fine-tunes or their merges. Consequently, this paper focuses on full model fine-tuning.

**Sparse finetuning.** An orthogonal approach to DPP is sparse fine-tuning, which reduces the size of delta parameters by modifying the fine-tuning process itself. This is achieved through techniques such as iterative masking and fine-tuning to create sparse DP (Guo et al., 2020; Liao et al., 2023; Fu et al., 2023). In contrast, DPP methods like DARE Yu et al. (2023a) are primarily post-hoc procedures that focus on pruning the delta weights of models that have already been fine-tuned, making DPP particularly valuable for the many fine-tuned models available on platforms like Hugging Face.

**Follow-up work on DARE.** The recent development of the DARE (Drops delta parameters And REscales) technique has significantly advanced the efficiency of finetuned-model pruning. This method sets most delta parameters to zero, maintaining the efficacy of Supervised Fine-Tuning without loss of performance. Highlighted in Goddard et al. (2024) within Arcee's MergeKit, a toolkit for merging large LMs, DARE has shown great potential enhancing model merging processes and has various practical applications. It has been successfully implemented in projects such as Prometheus Kim et al. (2024), an open-source LM for evaluating other LMs, and in medical LMs like Biomistral Labrak et al. (2024), which develops LMs for medical applications. The technique also supports specialized domains, as seen in Akiba et al. (2024), which focuses on a Japanese LLM with advanced reasoning capabilities. These implementations highlight the broad applicability and effectiveness of DARE in enhancing model merging strategies, thus making our improvements particularly relevant and practical.

## C.2 IMPLEMENTATION DETAILS

For decoder LLMs, following Yu et al. (2023a), we set the temperature to 0.0 for greedy decoding and limit the maximum number of generated tokens to 1,024 on GSM8K. For encoder-based LMs, we fine-tune BERT-base-uncased and RoBERTa-base for 10 epochs using a warmup strategy and a learning rate of 1e-4. Experiments are conducted on NVIDIA A100 GPUs.

## C.3 PROPOSED FRAMEWORK FOR DPP METHOD SELECTION.

We outline the procedure for selecting appropriate DPP methods in practice. As illustrated in Fig 4, if fine-tuning is not permitted and the data points (DPs) have large statistics, we recommend using WANDA for DPP (see Sections 5.3). If the DPs are not large, DAREx-q with $1/q_v$ should be applied when a validation set is available, otherwise, DAREx-q with $1/q_e$ is recommended (see Sec. 4.1). If the existing DPs are insufficient and fine-tuning is allowed, we suggest using AdamR-$L_2$ or $L_1$ with appropriate regularization weights to generate highly prunable DPs for DAREx-q and MP, respectively (see Sec. C.7). Among the two, AdamR-$L_2$+DAREx-q (DAREx-$L_2$-q) should be prefered due to flexibility and better performance as shown in Table 9 and Table 5.

## C.4 CONTROLLING PRUNING WITH REGULARIZAITON

In this section, we demonstrate the magnitude of regularization weight can control the degree of pruning that is required.

Fig 6 (a-d) illustrates the AdamR-$L_2$ fintuned RoBERTa model with different regularization weights on SST2, MRPC, COLA and STS-B datasets respectively and we use pruning rate $p \in [0, 0.9, 0.99, 0.999]$ to prune the delta parameters. To separate different pruning rates in the figure, we set log scaled pruning $-log(1-p)/3 = [0, 0.33, 0.67, 1.00]$ in x axis. With moderate regularization the model has a close

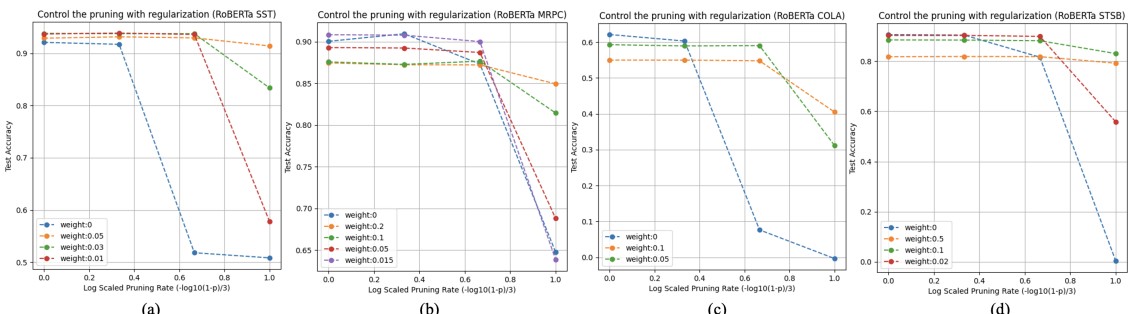

Figure 6: Control DARE pruning performance with $L_2$ regularizaiton on RoBERTa.(a-d) demonstrate results on SST2, MRPC, COLA, and STS-B, respectively. Aligning the degree of regularization proportionally to the pruning rate can lead to optimal performance.

performance to models without regularizaiton (weight 0) (even better *i.e.*weight 0.01 in Fig 6 (a)). This indicate adding moderate regularization won't hurt the model performance. To achieve moderate pruning performance $p = 0.99$, we can choose moderate regularization. For example, in fig 6 (c), choose weight 0.05 achieves the best performance when pruning rate $p = 0.99$ and outperforms the no regularization by more than 40% . When there is a need for extreme delta parameters pruning, it is recommended to increase the regularization weight. As shown in Figure 6 (d), weight 0.1 achieves the best performance when pruning rate $p = 0.999$ and outperforms no regualrizaiton by 80 %. It is notable with $L_2$ regualrization, all dataset can achieve 99.9% delta parameter pruning with minor performance drop. Finally, as shown in Fig 6 (d), a strong regularization weight 0.5 although face original performance drop, but demonstrate robustness (minor performance drop) to different level of pruning rate.

Fig 7 (a-d) depicts $L_2$ fine-tuned BERT models with varied regularization weights on SST2, MRPC, COLA, and STS-B datasets, respectively. Pruning rates $p \in [0, 0.9, 0.99, 0.999]$ are used. Log-scaled pruning $-log(1-p)/3 = [0, 0.33, 0.67, 1.00]$ are plot to separate different rates on the x-axis. Moderate regularization yields performance close to models without regularization (weight 0), sometimes even better (e.g., weight 0.05 in Fig 7 (a)), suggesting it won't hurt model performance. For moderate pruning ($p = 0.99$), moderate regularization suffices. For instance, in Fig 7 (a), weight 0.05 achieves optimal performance, outperforming no regularization by about 40%. For extreme pruning needs, increase the regularization weight. As in Fig 7 (b), weight 0.1 achieves peak performance at $p = 0.999$, surpassing no regularization by 60%. Notably, with AdamR-$L_2$, all datasets can achieve 99.9% DPP with minimal performance drop. Lastly, Fig 7 (c) shows that a strong weight of 0.1, despite an initial performance drop, exhibits robustness (minor drop) across different pruning rates.

## C.5 AdamR-$L_1$ finetuning

In this section, we focus on MP (Han et al., 2015a) and WANDA (Sun et al., 2023), utilizing AdamR-$L_1$ to enhance pruning performance for importance-based methods.

**MP:** We demonstrate AdamR-$L_1$ results in Table 7: the pruning rate consistently increased on datasets when applying magnitude pruning on $L_1$ fintuned models. This indicates AdamR-$L_1$ is an effective strategy in producing highly prunbale DPs for MP.

**WANDA:** As shown in Table 7, AdamR-$L_1$ improved WANDA's performance in most datasets. However, in the SST2 and STS-B datasets (Table 7), adding AdamR-$L_1$ negatively impacted pruning performance, as indicated by the numbers in red. This is because WANDA considers outlier activations Dettmers et al. (2022) in LLMs and uses $\Delta S_{ij} = |\Delta W_{ij}| \cdot |x_j|$ as its metric. Thus, some important DPs may be pruned

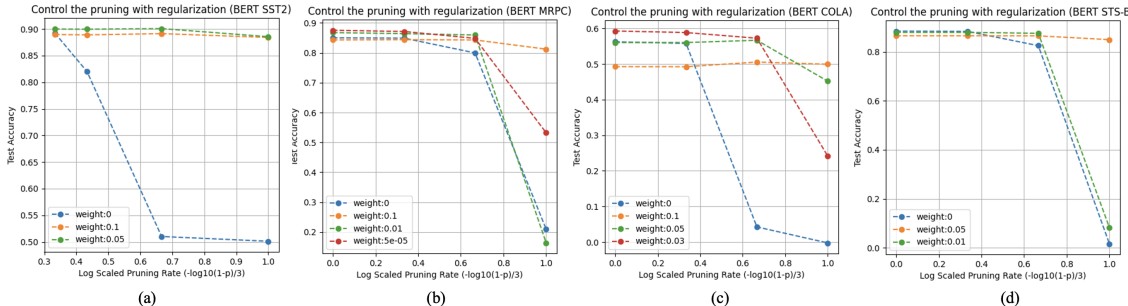

Figure 7: Control DARE pruning performance with $L_2$ regularization on BERT. (a-d) demonstrate results on SST2, MRPC, COLA, and STS-B, respectively. Aligning the degree of regularization proportionally to the pruning rate can lead to optimal performance.

Table 7: MP and WANDA with AdamR-$L_1$ on BERT and RoBERTa across various datasets

| Models | SST2 p=0.0 original | $L_1$ | p=0.90 original | $L_1$ | p=0.95 original | $L_1$ | p=0.99 original | $L_1$ | COLA p=0.0 original | $L_1$ | p=0.90 original | $L_1$ | p=0.95 original | $L_1$ | p=0.99 original | $L_1$ |
|---|---|---|---|---|---|---|---|---|---|---|---|---|---|---|---|---|
| BERT-MP | 90.25 | 89.91 | 91.40 | **89.68** | 84.29 | **89.22** | 51.15 | **83.14** | 56.24 | 56.52 | 37.47 | **47.49** | 21.58 | **44.59** | 8.49 | **12.30** |
| BERT-WANDA | | | 90.51 | 88.53 | 86.93 | 79.93 | 52.64 | **63.19** | | | 37.34 | **45.29** | 19.24 | **46.09** | 8.37 | **10.85** |
| RoBERTa-MP | 92.09 | 92.55 | 90.27 | **92.78** | 84.86 | **92.78** | 68.00 | **91.86** | 62.10 | 60.16 | 24.88 | **50.78** | 6.56 | **43.81** | 0.00 | **32.81** |
| RoBERTa-WANDA | | | 92.12 | 91.74 | 92.35 | 89.45 | 76.15 | 72.82 | | | 34.64 | **42.54** | 0.07 | **36.55** | 0.00 | **29.80** |

| Models | MRPC p=0.0 original | $L_1$ | p=0.90 original | $L_1$ | p=0.95 original | $L_1$ | p=0.99 original | $L_1$ | STS-B p=0.0 original | $L_1$ | p=0.90 original | $L_1$ | p=0.95 original | $L_1$ | p=0.99 original | $L_1$ |
|---|---|---|---|---|---|---|---|---|---|---|---|---|---|---|---|---|
| BERT-MP | 85.01 | 84.35 | 66.14 | **67.22** | 15.81 | **26.52** | 15.81 | 15.81 | 88.51 | 85.12 | 82.79 | **84.98** | 42.31 | **84.42** | 55.80 | **75.71** |
| BERT-WANDA | | | 72.85 | 64.22 | 22.75 | **23.49** | 15.81 | 15.81 | | | 80.71 | 76.11 | 50.70 | **70.80** | 40.92 | **54.21** |
| RoBERTa-MP | 90.05 | 90.07 | 76.39 | **85.17** | 75.17 | **78.93** | 74.80 | 74.80 | 90.57 | 89.87 | 82.70 | **87.43** | 74.58 | **86.25** | 21.26 | **58.53** |
| RoBERTa-WANDA | | | 82.21 | **84.68** | 76.24 | **78.86** | 74.80 | 74.80 | | | 84.56 | 78.13 | 76.17 | 65.42 | 20.91 | **25.87** |

when AdamR-$L_1$ is applied due to their small $|x_j|$. This suggests that important DPs are not always aligned with outlier activations, and that AdamR-$L_1$ may not be the optimal strategy for improving WANDA. For further analysis of outlier activations, see Appendix D.

## C.6 SCALE OF DELTA PARAMETERS

As demonstrated in Theorem 3.1, DARE is effective at pruning DPs when the mean and variance of $c_{ij}$ are small, and it generally outperforms importance-based methods in such conditions, as shown in Table 12. While most fine-tuned LLMs typically introduce small DPs Yao & Klimovic (2023); Lee et al. (2019), and $x$ is small, as indicated in Table 8, where $|\Delta W_{ij} x_j| \ll |\Delta W_{ij}|$, there are still instances where delta parameters are large. Specifically, Table 8 reports the magnitudes of $\Delta W$ and $c_{ij}$ across various LLMs. For instance, the MetaMath-LLaMA-7B Yu et al. (2023b), which fine-tunes LLaMA2-7B using two large datasets—MetaMathQA Yu et al. (2023b) and Proof-Pile-2 Azerbayev et al. (2023)—yields a mean delta parameter magnitude of 0.017, significantly larger than that of Abel-7B and MetaMath-7B. As illustrated in Table 6, DARE experiences a substantial performance drop when the pruning rate exceeds 0.1.

## C.7 ADAMR ALGORITHM

With a simple model, Sec. B demonstrated how $L_1$ and $L_2$ regularization can enhance DPP. We now show that when modifications to model finetuning are allowed, our approach can scale up to LLMs. In order to implement our regularization strategy, we propose replacing the weight decay of AdamW—the standard choice for training LLMs—with our custom regularization, as described in Sec. 5.2 and Sec. C.5. However, we do this carefully since Xie et al. (2024) found recently that weight decay can lead to large gradient norms

Table 8: Magnitude Statistics of $\Delta W_{ij}$ and $\Delta W_{ij}x_j$ of Llama2-7B fintuned LLMs on all dimensions and layers.

| Models | Magnitude | |
|---|---|---|
| | $\text{mean}(|\Delta W_{ij}|)$ | $\text{mean}(|\Delta W_{ij}x_j|)$ |
| Abel-7B | 7.3e-4 (4.3e-7) | 2.89e-5 (5.41e-9) |
| MetaMath-7B | 7.2e-4 (3.3e-7) | 2.85e-5 (6.02e-9) |
| Wizardmath-7B | 4.0e-4 (1.0e-7) | 1.56e-5 (1.67e-9) |
| MetaMath-llema-7B | 0.017 (1.9e-4) | 7.0e-4 (3.02 e-6) |

in the final training phase, deteriorating convergence. To address this issue, following Xie et al. (2024) we adjust our regularization strength based on the gradient norm. We call our new optimizer as AdamR. Note regularization here is based on the DP rather than simply on the weights of the finetuned model. The detailed algorithm, which we call AdamR, is presented in Algorithm 1.

---

**Algorithm 1** AdamR

**Input:** Pre-trained model $\boldsymbol{\theta}_p$,

1: $\boldsymbol{g}_t = \nabla L(\boldsymbol{\theta}_{t-1})$
2: $\boldsymbol{m}_t = \beta_1 \boldsymbol{m}_{t-1} + (1 - \beta_1)\boldsymbol{g}_t$
3: $\boldsymbol{v}_t = \beta_2 \boldsymbol{v}_{t-1} + (1 - \beta_2)\boldsymbol{g}_t^2$
4: $\hat{\boldsymbol{m}}_t = \frac{\boldsymbol{m}_t}{1-\beta_1^t}$
5: $\hat{\boldsymbol{v}}_t = \frac{\boldsymbol{v}_t}{1-\beta_2^t}$
6: $\bar{v}_t = \text{mean}(\hat{\boldsymbol{v}}_t)$
7: $\boldsymbol{\theta}_t = \boldsymbol{\theta}_{t-1} - \frac{\eta}{\sqrt{\hat{\boldsymbol{v}}_t}+\epsilon}\hat{\boldsymbol{m}}_t - \frac{\eta}{\sqrt{\bar{v}_t}+\epsilon}\lambda(\boldsymbol{\theta}_{t-1} - \boldsymbol{\theta}_p)$      ▷ $L_2$ regularization
8: $\boldsymbol{\theta}_t = \boldsymbol{\theta}_{t-1} - \frac{\eta}{\sqrt{\hat{\boldsymbol{v}}_t}+\epsilon}\hat{\boldsymbol{m}}_t - \frac{\eta}{\sqrt{\bar{v}_t}+\epsilon}\lambda\text{sign}(\boldsymbol{\theta}_{t-1} - \boldsymbol{\theta}_p)$      ▷ $L_1$ regularization

---

Recall from Sec. B that $L_2$ regularization promotes a small mean and variance of delta parameters, while $L_1$ regularization encourages sparse updates on important parameters, leading to larger variance. We also emphasize that our regularization here is based on the DP rather than simply on the weights of the finetuned model.

## C.8 DAREx-q algorithm

---

**Algorithm 2** Empirically find $q$

---

**Input:** Delta Weight $\boldsymbol{\Delta}_\theta$, Input Dataset $\mathcal{D}_I$, Pre-trained Model $\theta_P$, Pruning Rate $p$, Step Size $\Delta q$ $(\Delta \eta)$, Rounds $N$.

1: **initialization:** $error_{min} = \inf$
2: **for** $t \leftarrow 1 \rightarrow N$ **do**
3:     **if** Find global $q$ **then**
4:         $q_t = 1 - p + t \times \Delta q$                                     $\triangleright q \geq 1 - p$
5:         $\boldsymbol{\Delta}_\theta' = \mathcal{P}(\boldsymbol{\Delta}_\theta, q_t)$                         $\triangleright$ Prune and rescale by $q_t$
6:     **else**
7:         $\eta_t = t \times \Delta \eta$
8:         $\mathbf{q}_t = $ Algorithm 3$(\eta_t)$                   $\triangleright$ Resolve per-layer rescaling factor.
9:         $\boldsymbol{\Delta}_\theta' = \mathcal{P}(\boldsymbol{\Delta}_\theta, \mathbf{q}_t)$                      $\triangleright$ Prune and rescale by $\mathbf{q}_t$
10:     **end if**
11:     $\theta' = \theta_P + \boldsymbol{\Delta}_\theta'$                         $\triangleright$ Add pruned delta parameters to $\theta_P$
12:     **if** $\mathcal{D}_I$ is Validation Dataset **then**
13:         $error = \mathbb{E}_{(x,y) \in D_I} \mathbb{1}(f(\theta', x) \neq y)$               $\triangleright$ (1) Validation $q_v$
14:     **else**
15:         $error = \mathbb{E}_{x \in D_I} \text{abs}(f(\theta', x)^L - f(\theta_P + \boldsymbol{\Delta}_\theta, x)^L)$     $\triangleright$ (2) Last Layer's change $q_e$
16:     **end if**
17:     **if** $error \leq error_{min}$ **then**
18:         $error_{min} = error$ and $q_{best} = q_t$ ($\mathbf{q}_{best} = \mathbf{q}_t$)
19:     **end if**
20: **end for**
21: Return $q_{best}(\mathbf{q}_{best})$

---

---

**Algorithm 3** Analytically calculate $q$

---

**Input:** Delta Weight $\boldsymbol{\Delta}_\theta$, Input $x$, Pre-trained Model $\theta_P$, Pruning Rate $p$, Step Size $\Delta q$, Rounds $N \leq 1/\Delta q$, Empty $q$ List $\mathbf{q} = [\,]$, Probability $\gamma$, Constant $\eta$:

1: **initialization:** $x' = x$                                       $\triangleright$ Initialize layer input
2: **for** $\boldsymbol{\Delta}_\mathbf{W}, \mathbf{W}_P \in \boldsymbol{\Delta}_\theta, \theta_P$ **do**                        $\triangleright$ Loop each MLP layer
3:     **initialization:** $error_{min} = \inf$
4:     **for** $t \leftarrow 1 \rightarrow N$ **do**
5:         $q_t = 1 - p + t \times \Delta q$                                  $\triangleright q \geq 1 - p$
6:         $error = Eq\,(5)(\boldsymbol{\Delta}_\mathbf{W}, q_t, \gamma, x', \eta)$     $\triangleright$ Evaluate the objective in Eq (5) for the given $q = q_t$
7:         **if** $error \leq error_{min}$ and $t \neq N$ **then**
8:             $error_{min} = error$ and $q_{best} = q_t$
9:         **end if**
10:     **end for**
11:     $\mathbf{q}.append(q_{best})$                            $\triangleright$ Add best $q$ for current layer
12:     $x' = f(\boldsymbol{\Delta}_\mathbf{W} + \mathbf{W}_P, x)$       $\triangleright$ Calculate the output of the current layer of the fine-tuned model.
13: **end for**
14: Return $\mathbf{q}$

---

In this section, we propose and describe three strategies to determine the global rescaling factor. The per-layer rescaling factor follows a similar approach to the global one, with slight adjustments in determining the $\eta$ and

resolving $1/\mathbf{q}$ (see details in Algorithm 2 step 6.)

**Empirically finding** $1/q_v$ **using validation data:** In this approach, we use a validation dataset $\{x_v, y_v\} \in \mathcal{V}$ to empirically determine the best rescaling factor $q$. The goal is to solve the optimization problem $\arg\min_q \mathbb{E}_{\mathcal{V}}(f_q(x_v) = y_v)$, where $f_q$ represents the pruned model rescaled by $1/q$. The process is as follows:

- *Optimization Process:* We iterate over a range of $q$ values, adjusting the pruned model by rescaling with $1/q$. For each $q$, we measure the model's performance on the validation set by comparing the model's predictions $f_q(x_v)$ to the true labels $y_v$.

- *Outcome:* The $q$ value that results in the best performance on the validation set is selected as $1/q_v$. This $1/q_v$ is expected to provide the best overall performance across all layers when evaluated against the validation set.

- *Algorithm Details:* This process is detailed in Algorithm 2 with objective (1), which outlines the specific steps taken to find $1/q_v$.

**Empirically finding** $1/q_e$ **using output change:** This method builds on the core idea from Thm 3.1, aiming to empirically determine a global rescaling factor $q_e$ that balances the mean and variance in output embedding differences. Unlike the validation-based approach, this method does not rely on labeled data. Instead, it uses an unsupervised objective function based on the difference in the last layer's output embedding before and after pruning and rescaling. *Procedure:*

- *Data Usage:* We select a single fixed batch of data, which could be drawn from the training set or any natural dataset.

- *Inference:* The selected batch is passed through both the pruned and original models, and the output embeddings are compared.

- *Optimization:* The goal is to find the rescaling factor $q_e$ that minimizes the difference between these embeddings, ensuring that the rescaled pruned model closely approximates the original model's output.

*Efficiency:* This method is highly efficient as it only requires inference on a single batch of data, avoiding the need for a full validation set. *Algorithm Details:* The specific steps for this method are outlined in Algorithm 2 with objective (2), which provides a detailed procedure for determining $1/q_e$.

## C.9 ANALYTICAL PER-LAYER $1/\mathbf{q}$

The challenge in naively implementing a per-layer tuning of the rescaling factor is that if $q^\ell$ were selected to optimize, say, the mean output change $|h_i^{\mathrm{diff}}|$ averaged over all neurons in the $\ell$-th layer, this would require $L$ searches, which is computationally intractable. Instead, we turn to theory for a more efficient solution.

Our approach extends Thm.3.1 , which bounds $|h_i^{\mathrm{diff}}|$ for $q = 1 - p$, to obtain a high-probability bound for arbitrary $q$. This involves bounding both the mean (which is now non-zero) and the variance, then optimizing this bound over $q$ for each layer. In view of Thm. 3.1, it is unsurprising that the bound depends on the first- and second-order statistics of the coefficients $c_{ij}$. Consequently, the optimal $q$ can indeed vary per layer, depending on these statistics.

We need a high-probability bound that holds for all values of $q$, even as $q$ becomes small (on the order of, but still larger than, $1 - p$). In more detail, following the machinery of Thm. 3.1 to bound $|h_i^{\mathrm{diff}}|$, we apply Markov's inequality to the Laplace transform $\mathbb{E}[e^{\eta|h_i^{\mathrm{diff}}|}]$, $\eta > 0$. After some careful calculations (see App. E.2), for each fixed $\eta$, we can define $q^\ell(\eta)$ by solving a one-dimensional minimization problem involving $\eta$, $p$, and the statistics of $c_{ij}^\ell$ of the respective $\ell$-th layer (which are measurable). Specifically, this is given as

follows, for some probability of failure $\gamma \in (0, 1)$:

$$q(\eta) := \arg\min_q |\log \frac{2}{\gamma} + \eta \left(1 - \frac{1}{q}(1 - p)\right) \sum_j c_{ij} + \frac{\eta^2 \Phi(p) \sum_j c_{ij}^2}{4q^2}| \tag{5}$$

In typical concentration bounds, one selects the optimal $\eta$ by minimizing the upper bound obtained via Markov's inequality. However, this optimal $\eta$ is inversely proportional to $q^2$. For the small $q$ values we are interested in here (on the order of $1 - p$), this makes Markov's inequality loose. Instead, we propose selecting $\eta$ by a grid search over the mean output change of the last layer after rescaling with $q^L(\eta)$. This yields a value of $\eta = \eta_e$.

To give flexibility to the rescaling factors of other layers, allowing them to adjust based on the individual statistics of $c_{ij}$, while avoiding grid searches for all layers, we use the same value of $\eta = \eta_e$ and rescale each layer with $q^\ell(\eta_e)$. We denote the vector of selected per-layer rescaling factors as $1/\mathbf{q}_e = [1/q^1(\eta_e), \ldots, 1/q^L(\eta_e)]$. Additionally, by selecting $\eta$ by optimizing the performance on validation set rather than the mean output change (denote this $\eta_v$), we can arrive at an alternative implementation, $1/\mathbf{q}_v = [1/q^1(\eta_v), \ldots, 1/q^L(\eta_v)]$, when labeled validation data are available.

### C.10 STRUCTURAL PRUNING

Our random-based DAREx-q can also be utilized for structural DPP for enhanced hardware efficiency (Shen et al., 2022; Yao & Klimovic, 2023). In this setup, we randomly select $a\%$ of the input dimensions of a MLP layer and randomly retain $b\%$ of the DPs within the selected dimensions, achieving a total structured DP pruning rate of

Table 9: Structural Pruning on BERT.

| Dataset ($p = 0.99$) | SST2 | COLA |
|---|---|---|
| DARE | 51.00 (0.23) | 6.25 (1.41) |
| Struct-DAREx-q ($1/q_v$) | 83.40 (1.92) | 39.09 (1.50) |
| Struct-DAREx-$L_2$-q ($1/q_v$) | **87.64 (0.47)** | **53.71 (1.74)** |

$p = a \cdot b\%$. In Table 9 we use $a = 5, b = 20$, resulting in an equivalent pruning ratio $p = 0.99$. Our results show that DAREx-q significantly outperforms DARE, achieving over a 30% improvement. AdamR-$L_2$ unlocks the potential of random-based DPP for structural pruning, improving DARE without $L_2$ finetuning by over 40%.

### C.11 COMPARISON TO SPARSE FINETUNING

Sparse Finetuning (SFT) selects key weights by directly altering the fine-tuning process, using methods like iterative masking and fine-tuning to achieve sparse parameterization. In contrast, our DAREx-q are primarily post-hoc techniques that prune the weights of models after they have been fine-tuned.
We compare DPP with sparse fine-tuning Guo et al. (2020), evaluating both on the BERT-Large model. Consistent with Guo et al. (2020), we assess the performance of vanilla DARE and our DAREx-q with a rescaling factor of $1/q_v$, using a pruning rate of $p = 0.995$ (retaining only 0.5% of weights). As shown in Table 10, for MRPC and SST2, vanilla DARE performs slightly below the Diff-struct method, while DAREx-q remains competitive with Diff-struct and surpasses Diff. For STS-B and COLA, vanilla DARE underperforms, but our DAREx-q restores performance, making it competitive with Diff-struct and superior to Diff. Moreover, a key advantage of our DAREx-q is that it is a post-hoc method, allowing for easy application to pre-fine-tuned models, and is simpler to implement.

Next, we conduct an additional experiment to examine how our proposed method—combining the post-hoc DARE with AdamR-$L_2$ (DAREx-$L_2$)—compares to sparse fine-tuning. As shown in Table 11, for a pruning rate of $p = 0.999$, DAREx-$L_2$ outperforms the baseline across both datasets. It's important to note that, similar to sparse fine-tuning, AdamR-$L_2$ modifies the fine-tuning process. However, AdamR-$L_2$ only replaces

Table 10: Comparison DAREx-q with SFT

| $p = 0.995$ | MRPC | STS-B | COLA | SST2 |
|---|---|---|---|---|
| Full-tune | 91.0 | 86.9 | 61.2 | 93.2 |
| Diff | 87.0 | 83.5 | 60.5 | _92.5_ |
| Diff-struct | _89.7_ | **86.0** | **61.1** | **93.1** |
| DARE | 89.6 | 0.00 | 53.2 | 90.1 |
| DAREx-q $(1/q_v)$ | **89.9** | _84.2_ | _60.1_ | 92.2 |

Table 11: Comparison DAREx-$L_2$ with SFT

| $p = 0.999$ | MRPC | STS-B |
|---|---|---|
| Full-tune | 91.0 | 86.9 |
| Diff | 86.2 | 82.9 |
| Diff-struct | _88.2_ | _85.2_ |
| DARE | 70.2 | 0.00 |
| DAREx-$L_2$ (ours) | **88.5** | **86.3** |

Table 12: Similar to Table 2, **DAREx-q consistently provides significant gains for decoder models**.

| Dataset | GSM8K | | | | | | | |
|---|---|---|---|---|---|---|---|---|
| Methods | Abel-7B | | MetaMath-7B | | WizardMath-7B | | MetaMath-Qwen2-0.5B | |
| | $p = 0.95$ | $p = 0.99$ | $p = 0.95$ | $p = 0.99$ | $p = 0.95$ | $p = 0.99$ | $p = 0.95$ | $p = 0.99$ |
| No Pruning | 58.32 | | 65.50 | | 54.90 | | 42.00 | |
| WANDA | 18.20 | 0.00 | 23.58 | 0.00 | 21.50 | 0.00 | 0.00 | 0.00 |
| MP | 15.16 | 0.00 | 22.82 | 0.00 | 12.00 | 14.50 | 0.00 | 0.00 |
| DARE | 37.30 | 0.00 | 58.22 | 0.00 | 47.10 | 0.00 | 0.00 | 0.00 |
| DAREx-q $(1/q_v)$ | **47.20** | _20.20_ | 59.05 | **34.87** | _49.81_ | **35.63** | **30.70** | **19.17** |
| DAREx-q $(1/\mathbf{q}_v)$ | _44.50_ | 20.00 | _59.28_ | 32.06 | **50.64** | _34.34_ | 29.56 | 18.65 |
| DAREx-q $(1/q_e)$ | _42.99_ | **21.30** | **60.34** | _32.60_ | 49.05 | _33.97_ | _30.32_ | _18.95_ |
| DAREx-q $(1/\mathbf{q}_e)$ | 42.84 | 19.63 | _59.28_ | 28.05 | **50.64** | 33.58 | 28.80 | 17.66 |

the original AdamW optimizer while keeping the same task loss objective, making it a simpler implementation compared to methods like Diff, which require additional modifications.

### C.12 RANDOM-BASED METHODS GENERALLY OUTPERFORMS IMPORTANCE-BASED DPP

When DPs are not large, we show that random-based methods consistently outperform importance-based methods across all cases on decoder models, as presented in Table 12, with our DAREx-q achieving more than a 30% improvement.

### C.13 BROADER IMPACT

As discussed in Section 1 and demonstrated in prior work, DPP offers potential advantages for online serving, gradient communication in Federated Learning, and storage efficiency. Our systematic study of DPP methods and the proposed strategies that improve performance of existing methods can thus positively impact the usage of LLMs. We analyzed the impact of our approach on the efficiency of the machine learning system.

As stated in the introduction, a high pruning rate can benefit online serving, gradient communication in Federated Learning, and storage saving. To showcase this effectiveness, we compare the delta parameter loading time, the number of communication parameters, and the storage size at various pruning rates, as detailed in Table 13. Firstly, we found that the time to load the sparse compressed model decreases significantly as the pruning rate increases. In Federated Learning, only $p\%$ of the parameters need to be communicated, greatly enhancing communication efficiency. Additionally, when applied to storage saving, we further reduce memory usage by converting sparse tensors to CSR format, resulting in only 1.7 MB and 7.5 MB of storage with a 99.9% pruning rate for BERT and RoBERTa respectively. Consequently, pruning delta parameters can effectively enhance the efficiency of machine learning systems.

Table 13: Loading, Communication, Storage Efficiency with different pruning rate

| Models | Loading Time (s) | | | | Communication Size (# parameters) | | | | CSR Storage Size (MB) | | | |
|---|---|---|---|---|---|---|---|---|---|---|---|---|
| | pruning rate $p$ | | | | pruning rate $p$ | | | | pruning rate $p$ | | | |
| | No Pruning | 0.9 | 0.99 | 0.999 | No Pruning | 0.9 | 0.99 | 0.999 | No Pruning | 0.9 | 0.99 | 0.999 |
| BERT | 0.143 | 0.120 | 0.035 | 0.023 | $\approx 110$ M | $\approx 11$ M | $\approx 1.1$ M | $\approx 0.11$ M | 417.7 | 108.7 | 11.4 | 1.7 |
| RoBERTa | 0.165 | 0.125 | 0.038 | 0.026 | $\approx 125$M | $\approx 12.5$M | $\approx 1.25$M | $\approx 0.125$M | 475.6 | 102.3 | 16.2 | 7.5 |

Figure 8: Analysis Outlier Features and the $\Delta W x$ with SST2 fintuned RoBERTa. Mean (green line) means average of all feature dimensions. The $\Delta W x$ is extremely small on average.

## D  MASSIVE ACTIVATION AND OUTLIER FEATURES

Since Transformer-based models contain significantly large outliers that tend to concentrate in a few embedding dimensions Wei et al. (2022), and LLMs exhibit massive activations with high magnitude values on unique dimensions for specific tokens Sun et al. (2024), we analyze the influence of these large magnitude values in our study.

### D.1  OUTLIER FEATURE (ACTIVATION)

In this section, we empirically show the influence of outlier features. As shown in Table 7, WANDA outperforms MP on SST2, MRPC in original fintuned model, which indicates the validity of considering outlier features in DPP. Although outlier features will introduce larger activation inputs than normal features (Fig 8 (a)), with small delta parameter change, the mean and variance of $\Delta W_{ij} x_{ij}$ is still small (Fig 8 (b-c)) thus making DARE works well in delta parameter pruning. As a result, DARE results (Table 5 and Fig 7) outperforms that of WANDA and MP.

### D.2  MASSIVE ACTIVATION

The decoder based large language models (larger than 7-B) have massive activation Sun et al. (2024) that embrace large value on unique dimension on unique tokens (*i.e.*first token of a input). We leverage MetaMath-llema-7B Yu et al. (2023b) which finetune Llama2-7B with math datasets. We analyze each attention layer's input activations and identified the 2533-th,1415-th and 1512-th dimension as the massive activations, and shown in Fig 9 (a). It is notable 2533-th and 1415-th dimension is aligned with the pretrain model Llama2-7B's massive activation dimension Sun et al. (2024), but we have one new 1512-th dimension in MetaMath-llema-7B. When then studied the impact of layer normalization and found the massive activations is heavily downscaled by the layer normalizaiton's scale multiplier, which may fail to being considered as important by Wanda. We furtherly study the influence on removing the delta parameters that correspond to the those large manitude features by evaluating the results on GSM8K dataset. As shown in Table 14,

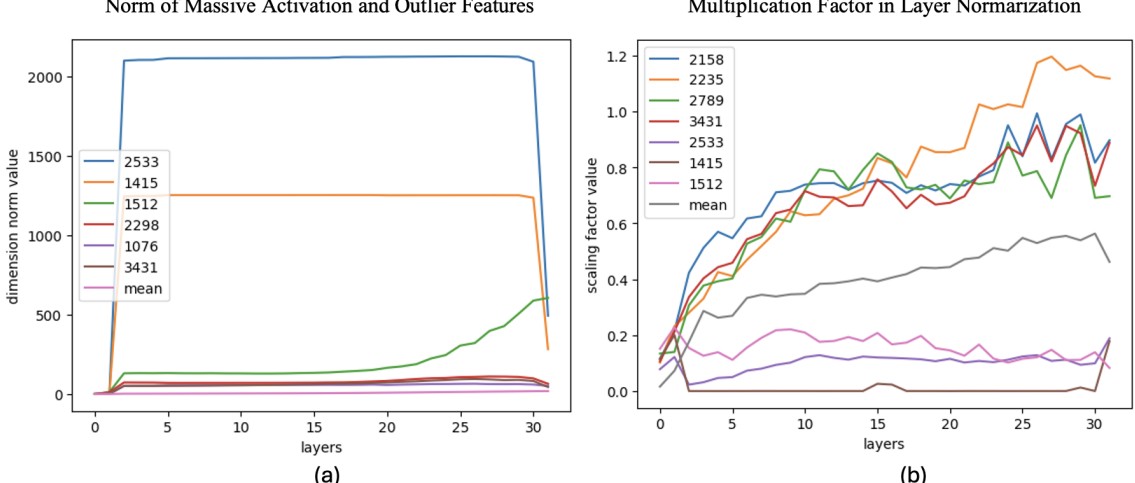

Figure 9: Analysis on Massive Activation and Outlier Features on MetaMath-llema-7B

removing the 1512-th massive activation features will bring significant performance drop. As a result, we suggest to maintain the delta parameters related to new massive activation features.

Table 14: Influnece of the delta parameters that correspond to large manitude features. Evaluation accuracy on GSM8K dataset.

| ID | Prune Status | Accuracy |
|------|---------------------|----------|
| None | No Pruning | 64.50 |
| 2533 | Massive Activations | 62.50 |
| 1512 | Massive Activations | 48.00 |
| 1415 | Massive Activations | 65.50 |
| 3431 | Outlier Feature | 61.00 |
| 2158 | Outlier Feature | 64.50 |
| 2350 | Outlier Feature | 62.50 |

## E  PROOFS

### E.1  PROOF OF THEOREM 3.1

The proof employs Hoeffding's inequality for subgaussian random variables. However, a straightforward application results in suboptimal dependence of the bound on the drop-rate $p$. Instead, we utilize a refinement of Hoeffding's bound tailored for (generalized) Bernoulli random variables, as developed by Kearns and Saul Kearns & Saul (2013). For large values of $p$, we adopt the improved bound proposed by Berend and Kontorovich Berend & Kontorovich (2013). These adaptations are detailed below.

Recall from Eq. (2) that the entries of $\boldsymbol{h}^{\text{diff}}$ are given as

$$h_i^{\text{diff}} = \sum_{j=1}^{n} \Delta W_{ij} x_j - \sum_{j=1}^{n} \frac{1}{1-p} \delta_{ij} \Delta W_{ij} x_j = \sum_{j=1}^{n} \left(1 - \frac{1}{1-p} \delta_{ij}\right) \Delta W_{ij} x_j,$$

where $\delta_{ij}$ are iid Bernoulli$(1 - \text{p})$ random variables. Fix any $i \in [m]$. Note $h_i^{\text{diff}}$ is weighted sum of $n$ iid random variables

$$A_{ij} := \left(1 - \frac{1}{1-p} \delta_{ij}\right).$$

We make the following observations about these random variables. First, a simple calculation gives that they are zero-mean, i.e. $\mathbb{E}[A_{ij}] = 0$ for all $j \in [n]$. Second, the random variables are bounded satisfying $-\frac{p}{1-p} \leq A_{ij} \leq 1$. Third, a simple calculations yields their variance $\mathbb{V}(A_{ij}) = p/(1-p)$.

Based on these, perhaps a first attempt in bounding $|h_i^{\text{diff}}|$ is applying Chebychev's inequality: For any $t > 0$,

$$\Pr(|h_i^{\text{diff}} - \mathbb{E}(h_i^{\text{diff}})| \geq t) \leq \frac{\mathbb{V}(h_i^{\text{diff}})}{t^2}.$$

Using the mean and variance calculations above, this yields with probability at least $1 - \gamma$:

$$|h_i^{\text{diff}}| \leq \frac{1}{\sqrt{\gamma}} \sqrt{\frac{p}{1-p}} \sqrt{\sum_{j=1}^{n} \Delta W_{ij}^2 x_j^2}.$$

The drawback is of course that this scales poorly with $\gamma$.

The natural way to improve the dependence of the bound on $\gamma$ is to use stronger concentration as materialized by a Chernoff bound. Since $A_{ij}$s are bounded, we can immediately apply Hoeffdings inequality for bounded random variables (Vershynin, 2020, Thm. 2.2.6) to find that for any $t > 0$:

$$\Pr\left(|h_i^{\text{diff}} - \mathbb{E}(h_i^{\text{diff}})| > t\right) \leq 2 \exp\left(-\frac{2t^2}{\frac{1}{(1-p)^2} \sum_{j \in [n]} c_{ij}^2}\right).$$

Therefore, for any $\gamma \in (0, 1)$ the following holds with probability at least $1 - \gamma$:

$$|h_i^{\text{diff}}| \leq \left(\frac{1/\sqrt{2}}{(1-p)}\right) \sqrt{\sum_{j \in [n]} c_{ij}^2} \sqrt{\log\left(\frac{2}{\gamma}\right)}.$$

Note the significant improvement over Chebychev's bound with respect to the scaling factor $\gamma$. However, the bound remains relatively loose in terms of $p$ for values of $p$ near the extremes of its range. To see this note that as $p \approx 0$, for which $A_{ij} \approx 0$ and $\mathbb{V}(A_{ij}) = p/(1-p) \approx 0$. Conversely, Hoeffding's bound, which scales as $1/(1-p)$, does not decrease as $p$ gets smaller. To get an improved bound, we can apply the Kearns-Saul ineqaulity (Kearns & Saul, 2013, Theorem 2), which is better suited for managing the characteristics of the distribution as $p$ approaches extremal values.

**Theorem E.1** (Kearns & Saul (2013)). *Let $R_j \in \{0, 1\}$ be iid* Bernoulli$(1 - \text{p}_j)$ *for $j \in [n]$. Let also constants $\alpha_j, j \in [n]$. Then, for all $t > 0$*

$$\Pr\left(\left|\sum_{j \in [n]} \alpha_j (R_j - (1 - p_j))\right| > t\right) \leq 2 \exp\left(-\frac{t^2}{\chi^2}\right), \tag{6}$$

*where $\chi^2 := \sum_{j \in [n]} \alpha_j^2 \Phi(p_j)$, and*

$$\Phi(x) = \frac{1 - 2x}{\log\left(\frac{1-x}{x}\right)}. \tag{7}$$

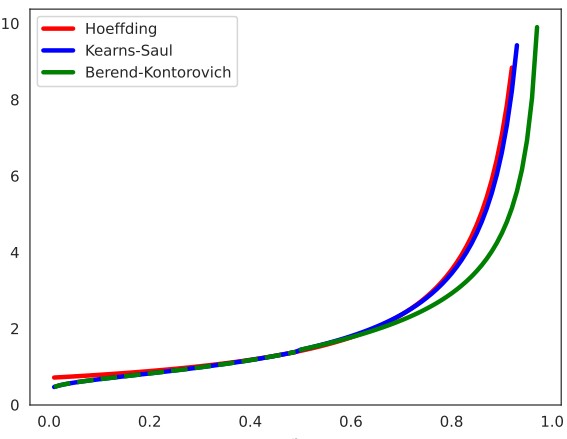

Figure 10: Visualization of the Hoeffding, Kearns-Saul and Berend-Kontorovich bounds. Note that the last is significantly improved over the vanilla Hoeffding bound for small/large values of $p$. Large values of $p$ corresponding to large drop-rates are particularly interesting in the DPP setting. Note also that all three bounds diverge as $p \to 1$. This is because of the rescaling factor $1/(1-p)$. However, note that the Berend-Kontorovich bound diverges significantly slower. This further signifies the importance of using this versus the vanilla Hoeffding bound to provide an explanation on why DARE performs well even for large values of $p$. See proof of Thm. 3.1 for details.

Applied to our setting, let $R_j \leftarrow \delta_{ij}$ and $\alpha_j \leftarrow \frac{1}{1-p} c_{ij}$. Then, the sum on the LHS of (6) is equal to $-\sum_{j \in [n]} A_{ij}$ and $\chi^2 = \frac{\Phi(p)}{(1-p)^2} \sum_{j \in [n]} c_{ij}^2$. Put together, this gives for any $t > 0$:

$$\Pr\left(|h_i^{\text{diff}}| > t\right) \le 2 \exp\left(-\frac{t^2}{\frac{1}{(1-p)^2} \Phi(p) \sum_{j \in [n]} c_{ij}^2}\right),$$

where $\Phi(p) := \frac{1-2p}{\log((1-p)/p)}$. Thus, for any $\gamma \in (0,1)$ the following holds with probability at least $1 - \gamma$:

$$|h_i^{\text{diff}}| \le \left(\frac{\sqrt{\Phi(p)}}{1-p}\right) \sqrt{\sum_{j \in [n]} c_{ij}^2} \sqrt{\log\left(\frac{2}{\gamma}\right)}. \tag{8}$$

To see why this improves upon Hoeffding's bound, note that the following hold for $\Phi(p)$:

1. $\Phi(p) \le 1/2$ for all $p \in [0,1]$
2. $\Phi(0) \to 0$ as $p \to 0$ or $p \to 1$.

The first property shows that the bound is strictly better than Hoeffding's for all values of $p$. The second property shows that the bound approaches 0 as $p \approx 0$, which is intuitive since in that case $A_{ij} \approx 0$ and $\mathbb{V}(A_{ij}) \approx 0$. On the other hand, for $p \to 1$, it is easy to show that $\sqrt{\Phi(p)}/(1-p) \to +\infty$. Hence, both Hoeffding and this improved bound predict that $|h_i^{\text{diff}}|$ can be large as $p \to 1$.

It turns out that the Kearns-Saul bound can be further improved for $p \ge 1/2$. Such an improvement is

important in our setting, since large drop-rates $p \gg 1/2$ are of particular interest in DPP. The improved inequality is due to (Berend & Kontorovich, 2013, Lemma 5). When applied to our setting, it improves the first term in to $\sqrt{\frac{2p}{p-1}}$.

To arrive now at the advertised statement of the theorem simply rewrite $\sum_{j \in [n]} c_{ij}^2$ in terms of the mean and variance of the influence statistics

$$\frac{1}{n} \sum_{j=1}^{n} \Delta W_{ij}^2 x_j^2 = \frac{1}{n} \sum_{j=1}^{n} c_{ij}^2 = \bar{c}_i^2 + \sigma_i^2.$$

This completes the proof of the theorem.

For completeness, we visualize the three bounds that we discussed above in Figure 10. The lower the better, thus note how the Kearns-Saul bound improves over naive Hoeffding and the Berend-Kontorovich bound improves over both for $p > 1/2$. In fact, this bound is tight as it can be seen from the central limit theorem as follows: From Eq. (2), the output change can be expressed as (dropping the subscript $i$ for convenience): $h = \sum_{j \in [n]} c_j (1 - 1/(1-p)\delta_j)$ where $\delta_j$ are iid Bernoulli$(1-p)$ RVs, we have dropped the subscript $i$ from Eq. (2) for simplicity and we have denoted $c_j = \Delta W_j x_j$. Our goal is to quantify the growth of the absolute output change $|h|$. Note that the random variables $Y_j = 1 - 1/(1-p)\delta_j$ have $E[Y_j] = 0$ and $Var[Y_j] = c_i^2 p/(1-p)$. Thus, by the central limit theorem, in the limit of large $n$, the (normalized) output change $(1/\sqrt{n})|h|$ is distributed as $\sqrt{p/(1-p)}\sqrt{\sum_i c_i^2}|G|$ where $G$ is a standard normal. Note the scaling $\sqrt{p/(1-p)} \sum_i c_i^2$ precisely matches the bound of Thm 3.1 for $p > 1/2$, demonstrating that it captures the true behavior of the output change with respect to $p$. The exponential high-probability bound of Thm 3.1 follows by a Hoeffding-type bound, but as demonstrated in Fig 10, to achieve this optimal p-scaling, we use a bound by Berend and Kontorovich that improves upon Hoeffding's bound on the moment generating function for generalized Bernoulli RVs.

### E.1.1 Lower bound

### E.2 Finding a rescaling factor balancing mean and variance

This section complements Sec. C.9 that describes the per-layer version of DAREx-q. Specifically, we show how to derive Eq. (5).

We study a more general version of DARE that rescales with a parameter $q$ that is not necessarily set to $1/(1-p)$. In this case, we have $A_{ij} = \left(1 - \frac{1}{q}\delta_{ij}\right)c_{ij}$, which we can rewrite as follows:

$$A_{ij} = \underbrace{\left(1 - \frac{1}{q}(1-p)\right)c_{ij}}_{=:\mu_{ij}} - \underbrace{\frac{1}{q}c_{ij}\left(\delta_{ij} - (1-p)\right)}_{=:R_j'}.$$

Following Kearns & Saul (2013), for any random variable $X$, it holds that $\Pr(X > 0) = \frac{1}{2}\mathbb{E}\left[1 + \frac{X}{|X|}\right]$. Observe that $\frac{1}{2}(1 + \frac{X}{|X|}) \leq e^{\eta X}$ for any positive number $\eta > 0, where \eta$ is a small positive number to ensure tightness of Eq. (9). Then, we obtain (alternatively, but equivalently, we can apply Markov's inequality to the

Laplace transform):

$$Pr[\sum_j A_{ij} > \epsilon] \leq \mathbb{E}\left[e^{\eta \sum_j \mu_{ij} - \frac{1}{q} c_{ij}(\delta_{ij} - (1-p)) - \eta\epsilon}\right] \tag{9}$$

$$= e^{-\eta\epsilon + \eta \sum_j \mu_{ij}} \mathbb{E}\left[\prod_j e^{-\frac{\eta}{q} c_{ij}(\delta_{ij} - (1-p))}\right] = e^{-\eta\epsilon + \eta \sum_j \mu_{ij}} \prod_j \mathbb{E}\left[e^{-\frac{\eta}{q} c_{ij}(\delta_{ij} - (1-p))}\right]$$

$$= e^{-\eta\epsilon + \eta \sum_j \mu_{ij}} \prod_j (pe^{\frac{\eta}{q} c_{ij}(1-p)} + (1-p)e^{-\frac{\eta}{q} c_{ij} p})$$

$$\leq e^{\eta\left(\sum_j \mu_{ij} - \epsilon\right) + \frac{\eta^2}{4} \sum_j \frac{c_{ij}^2}{q^2} \Phi(p)}, \text{(Kearns \& Saul, 2013, Lemma 1)}. \tag{10}$$

Since $\frac{c_{ij}^2}{q^2}\Phi(p) \geq 0$, choosing the optimal $\eta = \frac{-2\left(\sum_j \mu_{ij} - \epsilon\right)}{\frac{\Phi(p)}{q^2} \sum_j c_{ij}^2}$ minimizes Eq. (10). However, this may result in

a large $\eta$ especially for small $q$ that is of interest to us. When $\eta = \frac{-2\left(\sum_j \mu_{ij} - \epsilon\right)}{\frac{\Phi(p)}{q^2} \sum_j c_{ij}^2}$, for all $\epsilon > 0$, we have:

$$Pr[|\sum_j A_{ij}| > \epsilon] \leq 2e^{-\frac{q^2\left(\sum_j \mu_{ij} - \epsilon\right)^2}{\Phi(p) \sum_j c_{ij}^2}}. \tag{11}$$

Since $\mu_{ij} := \left(1 - \frac{1}{q}(1-p)\right) c_{ij}$, then with probability $1 - \gamma$, we have:

$$\epsilon(q) = \begin{cases} \left(1 - \frac{1}{q}(1-p)\right) \sum_j c_{ij} - \frac{1}{q}\sqrt{\log(\frac{2}{\gamma}) \sum_j c_{ij}^2 \Phi(p)}, & \text{if } \left(1 - \frac{1}{q}(1-p)\right) \sum_j c_{ij} - \epsilon > 0 \\ \left(1 - \frac{1}{q}(1-p)\right) \sum_j c_{ij} + \frac{1}{q}\sqrt{\log(\frac{2}{\gamma}) \Phi(p) \sum_j c_{ij}^2}, & \text{if } \left(1 - \frac{1}{q}(1-p)\right) \sum_j c_{ij} - \epsilon \leq 0 \end{cases}$$

Since $\eta \geq 0$, the second condition is always satisfied. Therefore,

- If $\frac{1}{q}\sqrt{\log\left(\frac{2}{\gamma}\right) \sum_j c_{ij}^2 \Phi(p)} = 0$, then $1 - p$ is simply the optimal rescaling.

- If $\frac{1}{q}\sqrt{\log\left(\frac{2}{\gamma}\right) \sum_j c_{ij}^2 \Phi(p)} > 0$, we can increase $q$ to compute the optimal scaling factor:

$$q = \arg\min_q \left(1 - \frac{1}{q}(1-p)\right) \sum_j c_{ij} + \frac{1}{q}\sqrt{\log(\frac{2}{\gamma}) \Phi(p) \sum_j c_{ij}^2}$$

$$= \arg\min_q \sum_j c_{ij} + \frac{1}{q}\left(\sqrt{\log(\frac{2}{\gamma}) \Phi(p) \sum_j c_{ij}^2} - (1-p) \sum_j c_{ij}\right)$$

Then the min value can be obtained by taking

$$q = 1 - p - \frac{\sqrt{\log(\frac{2}{\gamma}) \Phi(p) \sum_j c_{ij}^2}}{\sum_j c_{ij}}$$

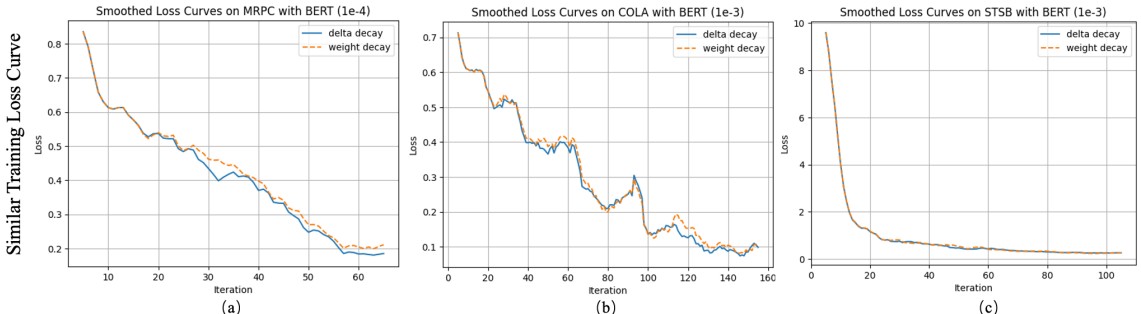

Figure 11: The training loss curves for delta decay and weight decay on the MRPC, COLA, and STSB datasets using BERT are compared. Both delta decay and weight decay exhibit similar training loss dynamics.

If $\sum_j c_{ij} > 0$ and $\sqrt{\log\left(\frac{2}{\gamma}\right)\Phi(p)\sum_j c_{ij}^2} > (1-p)\sum_j c_{ij}$, since the optimal $q > 0$, this suggests that increasing $q$ towards infinity is favorable. However, given $\eta = \frac{-2\left(\sum_j \mu_{ij}-\epsilon\right)}{\sum_j \frac{c_{ij}^2}{q^2}\Phi(p)}$, $\eta$ also tends to infinity, causing the loosening of Eq. (9). Thus, we propose selecting a fixed $\eta_\star$ to ensure the overall consistency of the inequalities in Eq. (9) and Eq. (10). Specifically, in our implementation of DAREx-q with per-layer tuning, $\eta_*$ is computed via grid search (see Sec. C.9). Substituting $\eta = \eta_\star$ into Eq. (10), we obtain:

$$\epsilon = \log\frac{2}{\gamma} + \eta_\star\left(1 - \frac{1}{q}(1-p)\right)\sum_j c_{ij} + \frac{\eta_\star^2\Phi(p)\sum_j c_{ij}^2}{4q^2}. \tag{12}$$

Given $\eta_\star$, we can find the optimal $q$ to minimize $|\epsilon|$ in Eq. (12), which arrives at Eq. (5).

## F  DAREx-$L_2$ TRAINING ANALYSIS

In this section, we analyze the training dynamics of AdamR, with a focus on DAREx-$L_2$. To align with the weight decay terminology and for simplicity, we refer to our DAREx-$L_2$ as "delta decay" in this section.

Concretely, we empirically inspect the training loss curve of delta decay and compare it with the (vanilla) weight decay. To be specific, we finetune a pretrained BERT-base model on MRPC, COLA and STSB datasets by using weight decay and delta decay respectively, then we plot the training loss curves for three fine-tuning tasks under different regularization strengths.

**Similar training dynamic to weight decay.** When using a reasonable regularization strength, as shown in Fig. 11, delta decay maintains stability and convergence on par with standard weight decay. This supports that delta decay having similar convergence and training dynamics.

**Large regularization.** We further increasing the regularization weight to $1e^{-2}$ and demonstrate the results in Fig 12. The results show that delta decay maintains stable training under a large regularization strength, whereas standard weight decay struggles and fails to converge.

## G  ADDITIONAL EXPERIMENTS ON SCALABILITY AND APPLICABILITY

In this section, we conducted extra experiments to show our method's strong scalability and applicability.

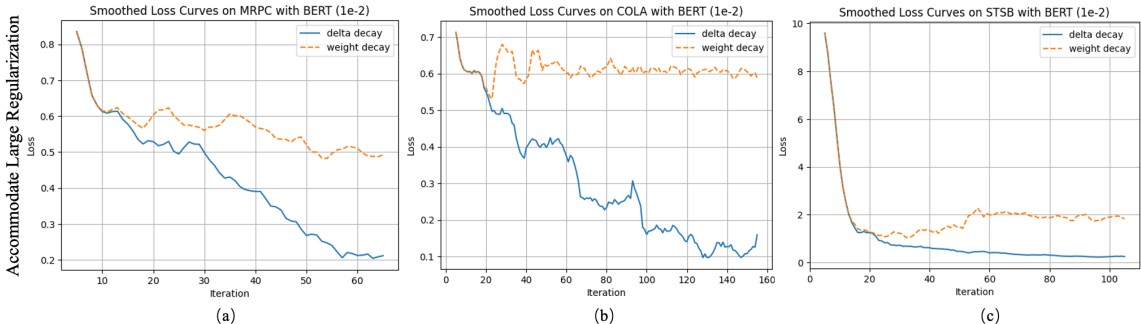

Figure 12: The training loss curves for delta decay and weight decay on the MRPC, COLA, and STSB datasets using BERT are analyzed under higher regularization strength. Delta decay is able to adapt to a larger regularization weight, whereas weight decay fails.

Table 15: Performance on CIFAR-100-ViT and Meta-Math-13B models.

| Dataset | CIFAR-100 | | GSM8K | |
|---|---|---|---|---|
| Methods | CIFAR-100-ViT | | MetaMath-13B | |
| | $p = 0.9$ | $p = 0.99$ | $p = 0.9$ | $p = 0.99$ |
| No Pruning | 91.98 | | 72.32 | |
| DARE | 90.53 | 32.74 | 67.09 | 0.00 |
| DAREx-q ($1/q_v$) | **91.68** | **84.11** | **68.84** | **44.73** |
| DAREx-$L_2$ | 91.01 | 83.84 | - | - |

### G.1 PERFORMANCE ON COMPUTER VISION TASK

We evaluated our method on computer vision tasks using a Vision Transformer (ViT) Dosovitskiy (2020) model pretrained on ImageNet-21k Ridnik et al. (2021) and fine-tuned on the CIFAR-100 dataset. As shown in Table 15, our approach consistently outperforms DARE across all scenarios, demonstrating substantial performance gains. Most notably, it achieves an improvement of over 50% at a pruning rate of 0.99 compared to baseline methods. These findings highlight the versatility of our method, showcasing its effectiveness not only in NLP tasks but also in the domain of computer vision.

### G.2 PERFORMANCE ON LARGER MODELS

We extended our method to the larger Llama2-13B model, fine-tuned on the MetaMath dataset Yu et al. (2023b), and assessed its performance on GSM8K. Despite computational constraints that limited us from fine-tuning the 13B model, DARq consistently outperformed other methods at both pruning rates (0.9 and 0.99), as detailed in Table 15. Notably, at a pruning rate of 0.99, DARq achieved an impressive performance improvement of over 44%. These results, alongside the consistent gains observed across different tasks (*e.g.*, computer vision and NLP) and model scales (*e.g.*, 0.5B, 7B and 13B), highlight the scalability and versatility of our method and demonstrate its potential for even larger models.

Table 16: Performance of AdamR-L2 on LoRA.

| $p = 0.99$ | SST2 | COLA | MRPC | STSB |
|---|---|---|---|---|
| DARE | 50.57 | 3.62 | 49.77 | 5.23 |
| AdamR-L2+DARE | **87.73** | **42.21** | **77.82** | **85.93** |

## G.3   LoRA with AdamR-$L_2$

This section presents the results of DARE on LoRA tuning and AdamR-L2 regularized LoRA tuning for SST2, COLA, MRPC, and STSB at a pruning rate of $p = 0.99$. As shown in Table 16, AdamR-L2 significantly improves DARE's performance across all tasks, achieving over a 30% improvement on each task and an impressive 80% gain on STSB.

