| DARq-$1/q_v$ | **47.20** | 20.20 | 59.05 | **34.87** | 49.81 | **35.63** | **30.70** | **19.17** |
| DARq-$1/\mathbf{q}_v$ | 44.50 | 20.00 | 59.28 | 32.06 | **50.64** | 34.34 | 29.56 | 18.65 |
| DARq-$1/q_e$ | 42.99 | **21.30** | **60.34** | 32.60 | 49.05 | 33.97 | 30.32 | 18.95 |
| DARq-$1/\mathbf{q}_e$ | 42.84 | 19.63 | 59.28 | 28.05 | **50.64** | 33.58 | 28.80 | 17.66 |

for a pruning rate of $p = 0.999$, AdamR-$L_2$+DARE outperforms the baseline across both datasets. It's important to note that, similar to sparse fine-tuning, AdamR-$L_2$+DARE modifies the fine-tuning process. However, AdamR-$L_2$ only replaces the original AdamW optimizer while keeping the same task loss objective, making it a simpler implementation compared to methods like Diff, which require additional modifications.

## C.12 RANDOM-BASED METHODS GENERALLY OUTPERFORMS IMPORTANCE-BASED DPP

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

Recall from Eq. (2) that the entries of $\boldsymbol{h}^{\mathrm{diff}}$ are given as

$$h_i^{\mathrm{diff}} = \sum_{j=1}^n \Delta W_{ij} x_j - \sum_{j=1}^n \frac{1}{1-p} \delta_{ij} \Delta W_{ij} x_j = \sum_{j=1}^n \left(1 - \frac{1}{1-p} \delta_{ij}\right) \Delta W_{ij} x_j,$$

where $\delta_{ij}$ are iid Bernoulli$(1-\mathrm{p})$ random variables. Fix any $i \in [m]$. Note $h_i^{\mathrm{diff}}$ is weighted sum of $n$ iid random variables

$$A_{ij} := \left(1 - \frac{1}{1-p} \delta_{ij}\right).$$

We make the following observations about these random variables. First, a simple calculation gives that they are zero-mean, i.e. $\mathbb{E}[A_{ij}] = 0$ for all $j \in [n]$. Second, the random variables are bounded satisfying $-\frac{p}{1-p} \le A_{ij} \le 1$. Third, a simple calculations yields their variance $\mathbb{V}(A_{ij}) = p/(1-p)$.

Based on these, perhaps a first attempt in bounding $|h_i^{\mathrm{diff}}|$ is applying Chebychev's inequality: For any $t > 0$,

$$\Pr(|h_i^{\mathrm{diff}} - \mathbb{E}(h_i^{\mathrm{diff}})| \ge t) \le \frac{\mathbb{V}(h_i^{\mathrm{