# OpenReview forum: "DARE the Extreme: Revisiting Delta-Parameter Pruning For Fine-Tuned Models"
_ICLR.cc/2025/Conference — ICLR 2025 Spotlight_

### Official Review · Reviewer_3h3b · 2024-11-03

**Soundness:** 4
**Presentation:** 3
**Contribution:** 4
**Rating:** 8
**Confidence:** 4

**Summary:**

The authors revisit delta-parameter pruning (DPP) for fine-tuned models, focusing on the limitations of the existing Random Drop and Rescale (DARE) method when dealing with high pruning rates or large delta parameters. They identify two key issues causing DARE's performance degradation: the excessively large rescaling factor at high pruning rates and the high mean and variance in the delta parameters. To address these challenges, they propose two algorithmic improvements:

1. **DARq**: A modification of DARE that adjusts the rescaling factor, leading to significant performance gains at high pruning rates.
2. **AdamR**: An in-training regularization technique that incorporates delta (L2) regularization during fine-tuning to control the mean and variance of delta parameters before applying DPP.

They demonstrate that their methods outperform the original DARE, especially at extreme pruning rates, and can be combined with parameter-efficient fine-tuning techniques like LoRA. Additionally, they show that importance-based pruning methods can outperform random-based methods when delta parameters are large.

The findings of this work seem very high utility and are of substantial increase in performance in edge cases where the original methods failed very strongly. A very good step forward for DPP methods.

**Strengths:**

- **Comprehensive Analysis**: The paper provides a thorough analysis of the limitations of the existing DARE method, identifying the root causes of its failure at high pruning rates.
- **Innovative Solutions**: The proposed methods, DARq and AdamR, are well-motivated and offer practical improvements over existing techniques.
- **Empirical Validation**: Extensive experiments on both encoder and decoder models across various datasets showcase the effectiveness of the proposed methods, with significant performance gains.
- **Flexibility and Compatibility**: The authors demonstrate that DARq can be combined with existing fine-tuning techniques like LoRA and can be used for structural pruning, increasing its practical utility.
- **Revisiting Importance-Based Pruning**: By exploring the application of importance-based pruning methods within DPP, they offer valuable insights into scenarios where these methods may be more effective.
- **Clear Recommendations**: The paper provides a practical pipeline for selecting the appropriate DPP method under different scenarios, which is valuable for practitioners.

**Weaknesses:**

- **Complexity of Methods**: While the proposed methods are effective, they introduce additional complexity, such as tuning the rescaling factor \( q \) in DARq, which may require validation data or additional computations.
- **Limited Theoretical Depth**: The theoretical analysis, while helpful, could be further deepened to provide more rigorous guarantees or insights into why the proposed methods work.
- **Applicability Scope**: The methods are primarily validated on language models and NLP tasks; it would strengthen the paper to demonstrate their applicability to other domains or types of neural networks.
- **Clarity in Presentation**: Some sections, particularly the methodology and theoretical analysis, could benefit from clearer explanations and more intuitive descriptions to make them accessible to a broader audience.
- **Dependence on Regularization**: The AdamR method relies on in-training modifications and specific regularization strategies, which may not always be feasible or may interact unpredictably with other training dynamics.

**Questions:**

1. **Scalability and Applicability to Other Domains**: While your methods have shown impressive results on language models, have you considered how DARq and AdamR might scale to even larger models or different architectures beyond NLP? Specifically, do you anticipate any challenges or necessary modifications when applying these techniques to other domains such as computer vision or speech recognition?

2. **Automated Selection of Rescaling Factor in DARq**: The selection of the rescaling factor \( q \) in DARq appears to be crucial for maintaining performance after pruning. Have you explored automated methods for selecting \( q \) without relying on a validation set or manual tuning? For instance, could analyzing the statistics of the delta parameters during pruning provide a way to determine an optimal \( q \) more systematically?

3. **Impact of AdamR on Training Dynamics**: Introducing delta regularization through AdamR alters the fine-tuning process. How does this affect the convergence, stability, and overall training dynamics compared to standard fine-tuning methods? Are there any trade-offs, such as increased training time or complexity, that practitioners should be aware of when integrating AdamR into their training pipelines?

---

> ### Author Response · Authors · 2024-11-21
> **Response Part I**
>
> We sincerely thank the reviewer for their positive and detailed feedback on our work. We are pleased that you found our contributions, including DARq and AdamR, to be innovative and impactful, and that our analysis, empirical validation, and practical recommendations were appreciated. Please see our responses below to your questions. We hope these address your concerns.
>
> -------------W1: Method Complexity---------
>
> **Answer**: Thank you for your question.
>
> As noted in the appendix (lines 1064–1066 in original submission, lines 1111-1113 in revision), tuning q is designed to be an efficient and lightweight process. It requires only a single batch of data and focuses on adjusting a single parameter, ensuring simplicity and ease of implementation. This approach minimizes both data handling and computational demands, making the overall process efficient.
>
> Furthermore, the associated pruning step is performed only once, which means its computational cost is fixed and does not impact the inference process. This minor complexity enables us to effectively minimize performance degradation for pruned models.
>
> --------- W3&Q1: Scalability and Applicability --------
>
> **Answer**: Thank you for your great suggestion.  We extended our experiments to evaluate the scalability and applicability of our method in the computer vision domain and with a larger 13B model.
>
> **Computer Vision**:  Following your suggestion, we applied our method to computer vision tasks using an ImageNet-21k pretrained Vision Transformer (ViT) model, fine-tuned on the CIFAR-100 dataset. As shown in the table below, our approach achieves a performance boost of over 50% at a pruning rate of 0.99, demonstrating its effectiveness beyond the NLP domain.
>
> **Larger 13-B Model**: We further evaluated our method on a larger Llama2-13B model fine-tuned on the MetaMath dataset [1] and tested its performance on GSM8K. Despite computational limitations that prevented us from fine-tuning this 13B model within the limited rebuttal phase, DARq consistently delivered superior pruning performance across both pruning rates (0.9 and 0.99). As evidenced in the results below, it achieved a performance boost of over 44% at a pruning rate of 0.99. The consistent improvements observed across various tasks and model sizes strongly suggest that our method is scalable to even larger models.
>
> | Model      | CIFAR-100-ViT |  | Meta-Math-13B | |
> |------------|--------------------------|---------------------------|--------------------------|---------------------------|
> |    |     p=0.9             |  p=0.99          |  p=0.9             |  p=0.99            |
> | No-Prune   | 91.98                   |      91.98               | 72.32                   |        72.32         |
> | DARE       | 90.53                   | 32.74                   | 67.09                  | 0                         |
> | DARq       | **91.68**                   | **84.11**                   | **68.84**                  | **44.73**                    |
> | AdamR-L2   | 91.01                   | 83.84                   | -                    | -                      |
>
> Thanks again for the suggestion. We have incorporated these new results into the Appendix G of our revision.
>
> [1] MetaMath: Bootstrap Your Own Mathematical Questions for Large Language Models
>
> ------------Q2: Automated Selection of Rescaling Factor in DARq and rely on a validation set .---------
>
> **Answer**: Thank you for raising this insightful question. We fully acknowledge the importance of automating the selection of the rescaling factor in DARq to reduce manual intervention.
>
> **Solutions without validation data**: As outlined in lines 253–255 of the original manuscript  (lines 255-258 in revised version), the optimal $q$ can be determined by monitoring the mean absolute output changes on a single batch of arbitrary text data. This method minimizes reliance on labeled validation datasets, enhancing its practical utility.
>
> **Feasibility of Auto Search Algorithm**:  Our results in Figure 1 show that both the output change and test performance have a convex/concave-like relationship with the inverse rescaling factor $q$.  This characteristic enables systematic search methods, such as the golden section search algorithm, which has logarithmic complexity $O(\log n)$, to efficiently identify the optimal $q$. We have adopted this approach in all our implementations to automate the selection process and will incorporate a detailed description of this automated selection process in the camera-ready version.
>
> We hope this helps address your concern and provides a clearer point regarding automation, and we appreciate your thoughtful feedback.

---

> ### Author Response · Authors · 2024-11-21
> **Response Part II**
>
> -----------W5: Dependence on Regularization. ----------
>
> **Answer:**  We believe that AdamR is a viable method for a wide range of fine-tuning tasks. Fundamentally, AdamR operates similarly to traditional weight decay; instead of penalizing deviations from zero, it penalizes deviations from the original weights. Given that weight decay is a standard practice in modern network training and fine-tuning, we extend this trust to AdamR. Furthermore, as evidenced by the results presented in Table 5, AdamR not only enhances the performance of pruned models but also shows promise in unpruned models in certain experiments. Additionally, the experimental results from previous answers on computer vision tasks reinforce the applicability of AdamR to other domains as well.
>
> -----------Q3: Impact of AdamR on Training Dynamics--------
>
> **Answer:**  Good question. We have not observed any stability or convergence issues in the experiments reported in the paper. Motivated by your question, we empirically inspect the training loss curve of AdamR and compare it with the standard training with weight decay.  To be specific, we finetune a pretrained BERT-base model on MRPC, COLA and STSB datasets by using weight decay and AdamR respectively, then we plot the training loss curves for three fine-tuning tasks under different regularization strengths.
>
> We report the results in Figures 11 and 12 in the revised manuscript and summarize the observations below:
>
> **Similar training dynamic to weight decay**:  When using a reasonable regularization strength, as shown in Fig. 11, AdamR maintains stability and convergence on par with standard weight decay. This supports that AdamR having similar convergence and training dynamics.
>
> **Accommodate Large regularization**: We further use a large regularization weight and demonstrate the results in Fig 12. The results show that AdamR maintains stable training under a large regularization strength, whereas standard weight decay struggles and fails to converge.
>
> We hope these additional results address your concern effectively.

---

### Official Review · Reviewer_xpsx · 2024-11-04

**Soundness:** 3
**Presentation:** 3
**Contribution:** 3
**Rating:** 8
**Confidence:** 4

**Summary:**

The paper addresses limitations in the random-rescaling Delta-parameter pruning method (DARE), which struggles with high pruning rates or large mean/variance in delta parameters. To tackle this, the authors propose two new approaches: (1) DARq, a post-training rescaling modification strategy, and (2) AdamR, a DDP-aware fine-tuning method. They validate the effectiveness of these methods through experiments on various model architectures and datasets.

**Strengths:**

* This work provides a clear analysis of why DARE struggles with high pruning rates and large delta parameter mean/variance cases. Additionally, it includes a theoretical proof of DARq’s effectiveness as a modified rescaling strategy.
* The paper presents comprehensive experiments on both encoder- and decoder-based language models across multiple datasets. DARq, with various rescaling variations, consistently outperforms other state-of-the-art methods across models and datasets, demonstrating its robustness. Moreover, DARq can be combined with parameter-efficient tuning approaches, such as LoRA, broadening its applicability.
* The AdamR fine-tuning method achieves promising results, effectively maintaining performance even at high pruning rates.

**Weaknesses:**

* This paper only compares their method with DARE, but there are lots of other methods proposed in the neural network pruning community. I wonder how this method compares to some existing pruning methods. For example, measuring the importance of weights in the original model and pruning the delta parameters based on that.

**Questions:**

* In Table 3, could you elaborate on why global rescaling outperforms local rescaling?
* In Table 5, after a 0.99 pruning rate, the pruned model seems to be able to achieve better performance than the original and AdamR-L2 fine-tuned models. For instance, in the RoBERTa-COLA cell, it reaches 59.01 at a 0.99 pruning rate with a standard deviation of 1.26, indicating that it can outperform the unpruned model (59.27) in optimal test runs. Does this suggest the potential for further pruning? Could you experiment with even higher pruning rates to assess the method's limits?

---

> ### Author Response · Authors · 2024-11-21
>
> We thank the reviewer for your time and highlighting our analysis of DARE’s limitations, the theoretical proof, and the robustness and applicability of DARq and AdamR across various models and datasets. Please see our responses below, which we hope address your concerns.
>
> --------W1: Compare with importance-based methods. ---------
>
> **Answer**: Great question: this was actually one of our own motivating questions. We have indeed adapted importance-based methods (such as magnitude pruning and the more recently introduced WANDA) for delta parameter pruning (see App. A). We discuss Importance-based DPP in Sec. 3.3 (lines 217-221 in original/revised submission) with detailed experimental results in the appendix (Tables 7, and 12).
> Overall, our methods significantly outperform importance-based DPP methods. To illustrate this, we summarize the results for the BERT model at a pruning rate of p=0.99 in the table below:
>
> | Model       | SST2   | COLA   | MRPC   | STS-B  |
> |-------------|--------|--------|--------|--------|
> | MP          | 51.15  | 8.49   | 15.81  | 55.80  |
> | WANDA       | 52.64  | 8.37   | 15.81  | 40.92  |
> | DARE        | 51.00  | 4.25   | 79.96  | 82.56  |
> | DARq-1/q_v  | 85.64  | 48.96  | 83.92  | 86.60  |
> | DARq-1/**q**_v  | 89.60  | 53.12  | 83.86  | 87.55  |
> | AdamR-L2    | 88.17 | 57.24  | 84.30  | 87.62  |
>
> As shown in the table, our methods outperform importance-based methods, such as MP and WANDA, by a significant margin.
>
> Similarly, for decoder models (Table 12 in Appendix): at p=0.95 on GSM8K, DARq outperforms WANDA and MP by over 30% across all models. At p=0.99, importance-based methods fail dramatically while our approach maintains strong performance.
>
> -----------Q1: Elaboration on Table 3---------
>
> **Answer**: Thank you for your question. Our experimental findings show that while local rescaling consistently outperforms global rescaling for encoder models (as noted in Table 2), the situation is more nuanced for decoder models. In decoder models, the performance gap between methods is relatively small (on average ~0.7% difference for q_v), with local rescaling maintaining an advantage in certain cases.
>
> Based on these findings, we recommend global rescaling for decoder models as stated in lines 358-359 in original submission (lines 359-360 in revision), since it offers implementation simplicity and can outperform local rescaling, while maintaining minimal performance gap when it underperforms.
>
> While we conducted extensive experiments to document these comparative performances, we acknowledge that the theoretical explanation for this encoder-decoder difference remains an open question for future research, and this is partly why we prioritized providing comprehensive empirical evidence to guide practical implementation choices.
>
> -------------Q2: Higher pruning rate for AdamR--------
>
> **Answer**: Thank you for your suggestion! Yes, AdamR-L2 enables effective pruning even at extremely high rates, as evidenced in Figures 6 and 7, where it supports delta parameter pruning rates of up to 0.999. Specifically, for the RoBERTa-COLA cell you referenced, Figure 6(c) (green line) illustrates a performance of 31.07 at this pruning rate, highlighting its robustness even under highly aggressive pruning scenarios.

---

> > ### Comment · Reviewer_xpsx · 2024-11-26
> >
> > I want to thank the authors for taking time to run the experiments to resolve my questions. I will change my score.

---

> > > ### Author Response · Authors · 2024-11-26
> > > **Thank you**
> > >
> > > Thank you for taking the time to review our rebuttal and for your thoughtful feedback. We’re delighted that our response has addressed your concerns. We truly appreciate your engagement and your increased score. Your encouragement means a lot to us—thank you!

---

### Official Review · Reviewer_CUxB · 2024-11-04

**Soundness:** 2
**Presentation:** 3
**Contribution:** 3
**Rating:** 8
**Confidence:** 3

**Summary:**

This work makes progress over DARE to obtain Prunable Delta Paramters at higher pruning rates. Towards this, the authors propose DARq, that modifies, the rescaling factor based on the insight that it is more prudent to look at output layer perturbations when pruning DPs as opposed to worrying about the expected change in delta parameters. The authors also propose to use AdamR, different from AdamW since it uses regularisation based on gradient norm, to obtain DPs that are more prunable than those obtained by AdamW. The authors also experiment with importance based pruning methods to show that they can obtain better result than random pruning. They also show that DARq works with structural pruning, LoRA.

**Strengths:**

S1. The problem considered is quite interesting and not well-addressed in the literature

S2. The paper is well written

S3. Empirical results evidently show that DARq is performs quite well. The other insights provided are valuable too.

**Weaknesses:**

W1. This paper seems quite exploratory. The authors do not conclude by providing a single method that they recommend to use for obtaining DPPs.

W2. Conclusions derived from Theorem 3.1 seem misleading. The authors provide an upper bound as opposed to a lower bound. On line 229, authors claim "from Thm. 3.1 as p increases, the absolute difference in outputs $|h^{diff}_{i} |$ grows at a rate of $\mathcal{O}((1 − p)^{-\frac{1}{2}})$, which is incorrect. But rather, $|h^{diff}_{i} |$ grows at most by $\mathcal{O}((1 − p)^{-\frac{1}{2}})$. It's not clear how tight this bound is to precisely make subsequent analysis hold true.



Typos:
Line 137: powerful

**Questions:**

Please refer to the weaknesses.

---

> ### Author Response · Authors · 2024-11-21
>
> Thank you for your time and thoughtful comments. We appreciate your interest in the good performance of DARq and the additional insights our paper brings. Please see our responses to your questions below. We hope these address your concerns and look forward to your updated feedback! We are happy to address any additional questions that you may have.
>
> --------W1: Exploratory and Recommendation--------
>
> **Answer**: Thank you for your comment, and we would like to take this opportunity to clarify that we have indeed provided recommendations.
>
> **Our recommendation** : We have made recommendations and presented a flowchart in Figure 4 to assist practitioners. Specifically, this flowchart delivers a clear message that one can customize the pruning strategies to meet different requirements and needs (e.g., scenarios with/without fine-tuning or validation data). Instead of a single solution, we believe our recommendations *better* align with the varied needs of practical applications. The utility and flexibility of this framework have been recognized in other reviewers’ comments (e.g., reviewers S9Bb and 3h3b), affirming its value in addressing the diverse needs of practical applications.
>
> **Comprehensive analysis**: Also, we respectfully believe that ‘exploratory’ is not a bad thing, particularly when it provides a unifying/broader viewpoint on a topic that, as the reviewer says, is of interest and not yet well-addressed in the literature. We aimed to approach the problem with a wide perspective that kicks off from the performance of vanilla DARE and identifies its limitations. We address these limitations by considering the circumstances of application (e.g., whether fine-tuning is possible) while investigating other candidate approaches to Delta Parameter pruning (e.g., importance-based methods).  The study provides concrete insights, which have also been acknowledged by other reviewers including S9Bb, xpsx, and 3h3b.
>
> -------- W2: Tightness of our bound ---------
>
> **Answer**:  Thank you for your comment, we respectfully disagree with this assessment. The upper bound in Theorem 3.1 is indeed tight with respect to p, which can be seen as follows:
>
> From Eqn (2), the output change can be expressed as:
> $$h = \sum_{j\in[n]} c_j (1 - 1/(1-p) \delta_j)$$
> where $\delta_j$ are iid $\text{Bernoulli}(1-p)$ RVs, we have dropped the subscript $i$ from Eqn (2) for simplicity and we have denoted $c_j=\Delta W_j x_j$. Our goal is to quantify the growth of the absolute output change $|h|$. Note that the random variables $Y_j=   c_j (1 - 1/(1-p) \delta_j)$ have $E[Y_j]=0$ and $Var[Y_j]=c_i^2p/(1-p)$. Thus, by the central limit theorem, in the limit of large $n$, the (normalized) output change $(1/\sqrt{n})|h|$ is distributed as $\sqrt{p/(1-p)}\sqrt{\sum_i c_i^2}|G|$ where $G$ is a standard normal. Note the scaling $\sqrt{p/(1-p) \sum_i c_i^2}$ precisely matches the bound of Thm 3.1 for $p>1/2$, demonstrating that it captures the true behavior of the output change with respect to $p$. The exponential high-probability bound of Thm. 3.1 follows by a Hoeffding-type bound, but as explained in App. E.1, to achieve this optimal p-scaling, we use a bound by Berend and Kontorovich that improves upon Hoeffding’s bound on the moment generating function for generalized Bernoulli RVs.
>
> We acknowledge the reviewer's point about precise language and have revised line 229 in revised submission to include "at most" for technical precision.
>
> Finally, we emphasize that beyond the theoretical foundation of Thm 3.1, which successfully motivates our improved DARq rescaling, our extensive experiments on encoder- and decoder-based language models across multiple datasets demonstrate that DARq consistently outperforms state-of-the-art methods.
>
> ----------W3: Typo--------
>
> **Answer** Thank you for pointing out this typo, we have corrected it in our revised version.
>
>
> Thank you again for your time!

---

> > ### Comment · Reviewer_CUxB · 2024-11-25
> > **Thank you for your responses**
> >
> > I wanted to thank the authors for their time during the rebuttal. Per the rebuttal on the comments made by all the reviewers, my concerns have been addressed. I will update my score.

---

> > > ### Author Response · Authors · 2024-11-25
> > > **Thank you**
> > >
> > > Thank you for taking the time to review our rebuttal. We sincerely appreciate your engagement and are delighted to hear that our response has addressed your concerns. Your updated score is greatly valued, and we are grateful for your constructive feedback throughout this process.

---

### Official Review · Reviewer_S9Bb · 2024-11-04

**Soundness:** 3
**Presentation:** 3
**Contribution:** 3
**Rating:** 6
**Confidence:** 3

**Summary:**

The paper proposes improved algorithms for Delta-Parameter Pruning (DPP), notably DARq and AdamR, which achieve substantial performance gains at high pruning rates.   It begins with a comprehensive analysis of DARE, an existing DPP approach, and leverages insights from this analysis to develop two enhanced algorithms. Furthermore, the proposed DARq method can be integrated with a parameter-efficient fine-tuning strategy to further minimize delta parameters. Additionally, the paper provides guidance on applying DPP across various scenarios.

**Strengths:**

1 The  paper provides a detailed analysis of DARE about the failure cases

2 base on the observation of DARE, they proposed two improved methods

3 their present the effectiveness of their method through theoretical and experimental points.

4 good presentation for reading

5 consider different scenarios

**Weaknesses:**

see questions

**Questions:**

1 could you explain why you present a different range of regulation strength instead of a unity strength range to better understand the trend of the scale of dp in the Figure 6 ?

2 how do you conduct the experiments on the different regulation strengths in Figure 4? do they keep all other hyperparameters the same?

3 have you try lora + adamr-l2/adamr-l1?

---

> ### Author Response · Authors · 2024-11-21
>
> We thank the reviewer for the thoughtful feedback and for highlighting our detailed analysis of DARE, the development of two improved methods, strong theoretical and experimental validation, clear presentation, and consideration of different scenarios. Please see our responses below, which we hope will effectively address your concerns.
>
> ----Q1: Rationale for using different regularization strength instead of a unity strength -------
>
> **Answer**: Thank you for your question. As mentioned in lines 862-867 in original submission (lines 909-916 in revised version), the different ranges of regularization strength were chosen to align with specific pruning rate requirements, allowing us to optimize performance for both moderate and extreme pruning levels.
> To elaborate, in cases of moderate pruning (e.g., p=0.99), a moderate regularization weight provides optimal results. For example, in Figure 6(c), a regularization weight of 0.05 achieves the highest performance at this pruning rate. However, for more extreme pruning demands, such as p=0.999, a stronger regularization weight is necessary to maintain performance. As shown in Figure 6 (d), a weight of 0.1 not only achieves the best performance for p=0.999 but also outperforms the non-regularized baseline by approximately 80%.
>
> ----Q2 : Experimental Setup----
>
> **Answer**: Thank you for your question.  We suspect that you are referring to the different regularization strengths shown in Figure 6, as Figure 4 is the flowchart; please correct us otherwise. Assuming that, then: Yes, in our experiments examining the effects of regularization strength, we kept all other hyperparameters the same, adjusting only the regularization weight.
>
> ---Q3: Combining LoRA with AdamR-----
>
> **Answer**: Thank you for your question. Following your suggestion, we conducted experiments on LoRA with AdamR-$L_2$ and present the results of DARE applied to both standard LoRA tuning and AdamR-$L_2$ regularized LoRA tuning for SST2, COLA, MRPC, and STSB at a pruning rate of $p=0.99$.
>
> | p=0.99 | SST2   | COLA   | MRPC   | STSB   |
> |-------------|--------|--------|--------|--------|
> | no-reg      | 50.57 | 3.62 | 49.77 | 5.23 |
> | AdamR-L2    | **87.73** | **42.21** | **77.82** | **85.93** |
>
> It is shown that the AdamR-$L_2$ improved the performance. Thanks again for the suggestion. We have incorporated these new results into the Appendix G of our revision.

---

> > ### Comment · Reviewer_S9Bb · 2024-11-25
> >
> > I would like to thank the authors for their efforts in addressing the majority of my concerns and questions. I am pleased with the revisions and would be happy to maintain the positive score as the final rating.

---

> > > ### Author Response · Authors · 2024-11-26
> > > **Thank you**
> > >
> > > Thank you for your time and support throughout this process. We’re pleased our revisions addressed your concerns and truly appreciate your positive score—thank you!

---

### Comment · Area_Chair_Sb4K · 2024-11-22
**Discussion**

Dear reviewers,

The authors have responded to your reviews.

Until November 26th @ 2359 (AOE time) reviewers and authors can freely exchange responses, so if there any clarifications you require from the authors, now is the time to seek them!

Best,

AC

---

### Meta-Review · Area_Chair_Sb4K · 2024-12-13

**Metareview:**

This paper analyses the "DARE" delta-parameter pruning method, and its failure mechanisms. It then proposes two algorithmic improvements in the form of "DARq" and "AdamR". Reviewers appreciated the motivation and the analysis of DARE, as well as strong, comprehensive experimental results. Criticisms were fairly minor, but included a lack of comparison to other pruning methods, issues with the theory, and a lack of validation outside of language domains (points that were later addressed by the authors). The authors did well with their responses, leading to two reviewers increasing their scores. With entirely positive reviews, solid experiments, and a lack of major weaknesses, I believe this paper should be accepted.

**Additional Comments On Reviewer Discussion:**

Reviewer CUxB had concerns about the paper being too exploratory, and Theorem 3.1 which were both successfully addressed by the authors leading to a score increase. Reviewer xpsx wanted to see comparisons to other pruning techniques that were then provided by the authors, leading to another score increase. The issues outlined being successfully addressed contributed positively to my decision.

---

### Decision · Program_Chairs · 2025-01-22

Accept (Spotlight)